# Absolute quantitation of individual SARS-CoV-2 RNA molecules provides a new paradigm for infection dynamics and variant differences

Jeffrey Y Lee[1†], Peter AC Wing[2,3†], Dalia S Gala[1], Marko Noerenberg[1,4], Aino I Järvelin[1], Joshua Titlow[1], Xiaodong Zhuang[2], Natasha Palmalux[4], Louisa Iselin[1], Mary Kay Thompson[1], Richard M Parton[1], Maria Prange-Barczynska[2,5], Alan Wainman[6], Francisco J Salguero[7], Tammie Bishop[2,5], Daniel Agranoff[8], William James[6,9], Alfredo Castello[1,4]*, Jane A McKeating[2,3]*, Ilan Davis[1]*

[1]Department of Biochemistry, The University of Oxford, Oxford, United Kingdom; [2]Nuffield Department of Medicine, The University of Oxford, Oxford, United Kingdom; [3]Chinese Academy of Medical Sciences (CAMS) Oxford Institute (COI), The University of Oxford, Oxford, United Kingdom; [4]MRC-University of Glasgow Centre for Virus Research, The University of Glasgow, Glasgow, United Kingdom; [5]Ludwig Institute for Cancer Research, The University of Oxford, Oxford, United Kingdom; [6]Sir William Dunn School of Pathology, The University of Oxford, Oxford, United Kingdom; [7]UK Health Security Agency, UKHSA-Porton Down, Salisbury, United Kingdom; [8]Department of Infectious Diseases, University Hospitals Sussex NHS Foundation Trust, Brighton, United Kingdom; [9]James & Lillian Martin Centre, Sir William Dunn School of Pathology, The University of Oxford, Oxford, United Kingdom

*For correspondence:
alfredo.Castello@glasgow.ac.uk (AC);
jane.mckeating@ndm.ox.ac.uk (JAM);
ilan.davis@bioch.ox.ac.uk (ID)

†These authors contributed equally to this work

Competing interest: The authors declare that no competing interests exist.

**Abstract** Despite an unprecedented global research effort on SARS-CoV-2, early replication events remain poorly understood. Given the clinical importance of emergent viral variants with increased transmission, there is an urgent need to understand the early stages of viral replication and transcription. We used single-molecule fluorescence in situ hybridisation (smFISH) to quantify positive sense RNA genomes with 95% detection efficiency, while simultaneously visualising negative sense genomes, subgenomic RNAs, and viral proteins. Our absolute quantification of viral RNAs and replication factories revealed that SARS-CoV-2 genomic RNA is long-lived after entry, suggesting that it avoids degradation by cellular nucleases. Moreover, we observed that SARS-CoV-2 replication is highly variable between cells, with only a small cell population displaying high burden of viral RNA. Unexpectedly, the B.1.1.7 variant, first identified in the UK, exhibits significantly slower replication kinetics than the Victoria strain, suggesting a novel mechanism contributing to its higher transmissibility with important clinical implications.

## Editor's evaluation

This manuscript uses imaging and analysis techniques to study the kinetics and diversity of RNA production in SARS-CoV-2 infected cells. This presents a novel insight into the biology and life cycle of the virus, and this knowledge has the potential to inform future interventions for COVID-19.

## Introduction

Severe acute respiratory syndrome coronavirus 2 (SARS-CoV-2) is the causative agent of the COVID-19 pandemic. The viral genome consists of a single positive strand genomic RNA (+gRNA) approximately 30 kb in length that encodes a plethora of viral proteins (*Kim et al., 2020*; *Zhao et al., 2021*). SARS-CoV-2 primarily targets the respiratory tract and infection is mediated by Spike protein binding to human angiotensin-converting enzyme (ACE2), where the transmembrane protease serine 2 (TMPRSS2) triggers fusion of the viral and cell membranes (*Hoffmann et al., 2020*; *Wan et al., 2020*). Following virus entry and capsid trafficking to the endoplasmic reticulum, the first step in the replicative life cycle is the translation of the gRNA to synthesise the replicase complex. This complex synthesises the negative sense genomic strand, enabling the production of additional positive gRNA copies. In addition, a series of shorter subgenomic RNAs (sgRNAs) are synthesised that encode the structural matrix, Spike, nucleocapsid and envelope proteins, as well as a series of non-structural proteins (*Kim et al., 2020*; *Sola et al., 2015*). The intracellular localisations of these early events were described using electron microscopy (*Laue et al., 2021*) and by antibody-based imaging of viral double-stranded (ds)RNA (*Lean et al., 2020*). However, the J2 dsRNA antibody lacks sensitivity and specificity at early times post infection as the low abundance of SARS-CoV-2 dsRNA is indistinguishable from host dsRNAs (*Dhir et al., 2018*). Our current knowledge of these early steps in the SARS-CoV-2 replicative life cycle is poorly understood despite their essential role in the establishment of productive infection.

Since the initial outbreak in the Wuhan province of China in 2019, several geographically distinct variants of concern (VOCs) with altered transmission have arisen (*Chen et al., 2020*; *Lythgoe et al., 2021*). Emerging VOCs such as the recently named Alpha strain (previously known and referred to herein as B.1.1.7), first detected in Kent in the UK, possess a fitness advantage in terms of their ability to transmit compared to the Victoria (VIC) isolate, an early strain of SARS-CoV-2 first detected in Wuhan in China (*Caly et al., 2020*; *Davies et al., 2021*; *Kidd et al., 2021*; *Volz et al., 2021*). Many of the VOCs encode mutations in the Spike (S) protein (*Rees-Spear et al., 2021*) and, consequently, the effects of these amino acid substitutions on viral entry and immuno-evasion are under intense study (*Kissler et al., 2021*; *Washington et al., 2021*). However, some of the mutations map to non-structural proteins, so could impact viral replication dynamics. To date, the early replication events of SARS-CoV-2 variants have not been characterised as the current techniques for quantifying SARS-CoV-2 genomes and replication rates rely on bulk approaches or have limited sensitivity.

The use of single-molecule and single-cell analyses in biology offers unprecedented insights into the behaviour of individual cells and the stochastic nature of gene expression that are often masked by population-based studies (*Fraser and Kaern, 2009*; *Raj and van Oudenaarden, 2009*). These approaches have revealed how cells vary in their ability to support viral growth and how stochastic forces can inform our understanding of the infection process (*Billman et al., 2017*; *Boersma et al., 2020*; *Chou and Lionnet, 2018*; *Shulla and Randall, 2015*; *Singer et al., 2021*). Fluorescence in situ hybridisation (FISH) was previously used to detect RNAs in hepatitis C virus and Sindbis virus-infected cells with high sensitivity (*Garcia-Moreno et al., 2019*; *Ramanan et al., 2016*; *Singer et al., 2021*). This approach has been applied to SARS-CoV-2 in a limited capacity (*Burke et al., 2021*; *Rensen et al., 2021*) with most studies utilising amplification-based signal detection methods to visualise viral RNA (*Best Rocha et al., 2020*; *Carossino et al., 2020*; *Guerini-Rocco et al., 2020*; *Jiao et al., 2020*; *Kusmartseva et al., 2020*; *Lean et al., 2020*; *Liu et al., 2020*). These experiments used either chromogenic histochemical detection using bright field microscopy or detection of fluorescent dyes, which both lack the sensitivity to detect individual RNA molecules. Consequently, the kinetics of SARS-CoV-2 RNA replication and transcription during the early phase of infection are not well understood and lack quantitative, spatial and temporal information on the genesis of gRNA and sgRNAs. To address this gap, we developed a single-molecule (sm)FISH method based on earlier published protocols (*Femino et al., 1998*; *Raj et al., 2008*; *Singer et al., 2021*; *Titlow et al., 2018*) to visualise SARS-CoV-2 RNAs with high sensitivity and spatial precision, providing a powerful new approach to track infection through the detection and quantification of viral replication factories. Our results uncover a previously unrecognised heterogeneity among cells in supporting SARS-CoV-2 replication and a surprisingly slower replication rate of the B.1.1.7 variant when compared to the early lineage VIC strain.

## Results

### SARS-CoV-2 genomic RNA at single-molecule resolution

To explore the spatial and temporal aspects of SARS-CoV-2 replication at single-molecule and cell levels, we carried out smFISH experiments with fluorescently labelled probes directed against the 30 kb viral gRNA. 48 short antisense DNA oligonucleotide probes were designed to target the viral ORF1a and labelled with a single fluorescent dye to detect the positive sense gRNA, as described previously (*Gaspar et al., 2017*; *Figure 1A*). The probe set detected single molecules of gRNA within SARS-CoV-2-infected Vero E6 cells, visible as well-resolved diffraction-limited single spots with a consistent fluorescence intensity and shape (*Figure 1B*). Treatment of the infected cells with RNase or the viral polymerase inhibitor remdesivir (RDV) ablated the probe signal, confirming specificity (*Figure 1—figure supplement 1A*). To assess the efficiency and specificity of detection of the +ORF1a probe set, we divided the probes into two groups of 24 alternating oligonucleotides ('ODD' and 'EVEN') that were labelled with different fluorochromes. Interlacing the probes minimised chromatic aberration between spots detected by the two colours (*Figure 1C*). Analysis of the SARS-CoV-2 gRNA with these probes showed a mean distance of <250 nm between the two fluorescent spots, indicating near-perfect colour registration and a lack of chromatic aberration. 95% of the diffraction-limited spots within infected cells were dual labelled, demonstrating efficient detection of single SARS-CoV-2 gRNA molecules (*Figure 1C*). To assess whether virion-encapsulated RNA is accessible to the probes, we immobilised SARS-CoV-2 particles from our viral stocks on glass and incubated them with the +ORF1a probes. We observed a large number of spots in the immobilised virus preparation that was compatible with single RNA molecules (*Figure 1—figure supplement 1B*), suggesting that detection of RNA within viral particles was achieved.

To verify the specificity of the +ORF1a probes for SARS-CoV-2, we aligned their sequences against other coronaviruses and the transcriptomes of both human and African green monkeys. Many of the oligonucleotides showed mismatches with SARS-CoV-1, MERS, and other coronaviruses along with human and green monkey RNAs (*Figure 1D*). The specificity of the +ORF1a probes highlights the applicability of our probes to detect SARS-CoV-2 in different mammalian hosts. The high level of mismatches with other coronaviruses predicts that the +ORF1a probes are unlikely to hybridise with RNAs of other coronaviruses. To evaluate this, we assessed the ability of the +ORF1a probe set to hybridise RNA from the common cold coronavirus HCoV-229E. Although the antibody against dsRNA (J2) detected dsRNA foci in the HCoV-229E-infected Huh7.5 hepatoma cells, no signal was detected with the SARS-CoV-2 +ORF1a probe set (*Figure 1—figure supplement 1C*). In contrast, the +ORF1a probe set bound with intense signals in J2-positive SARS-CoV-2-infected Calu-3 cells.

Next, we tested whether smFISH could be used to detect SARS-CoV-2 RNAs in formalin-fixed and paraffin-embedded (FFPE) lung tissue from experimentally infected Golden Syrian hamsters. The animals were inoculated with SARS-CoV-2 BVIC01 ($5 \times 10^4$ plaque-forming unit [PFU]) by intranasal delivery and infection assessed by qPCR measurement of viral RNA (mean $6.5 \times 10^6$ copies/ml) and titration of infectious virus (mean $5.3 \times 10^3$ PFU/ml) in throat swabs at 4 days post infection. Animals were assessed daily for the onset of clinical symptoms with the most severe being the presence of laboured breathing that was noted in all infected animals from day 3 post infection onwards (*Dowall et al., 2021*). Infected animals lost weight from day 1 post infection and by day 4 a loss of 10% total body mass was recorded; however, no significant change was noted in body temperature. The +ORF1a probes detected SARS-CoV-2 gRNA in a representative section of the infected lung tissue, showing variable intracellular RNA levels within the lung tissue (*Figure 1E*). Our results demonstrate the specificity of the +ORF1a probe set to detect single molecules of SARS-CoV-2 gRNA within infected tissue samples fixed and treated in a manner similar to clinically derived tissues. We conclude that smFISH is likely to work well in clinical studies on material derived from infected patients and provide a highly sensitive method to visualise viral RNA in these samples.

Having established smFISH for the detection of SARS-CoV-2 gRNA, we used this technique to assess both the quantity and distribution of gRNA during infection. Vero E6 cells were inoculated with virus at a multiplicity of infection (MOI) of 1 for 2 hr and non-internalised viruses were removed by trypsin digestion to synchronise the infection. At 2 hr post infection (hpi), most fluorescent spots correspond to single gRNAs along with a small number of foci harbouring several gRNA copies (*Figure 1F*), consistent with early RNA replication events. By 8 hpi, we noted an expansion in the number of bright multi-gRNA foci, and at 24 hpi there was a further increase in the number of multi-RNA foci

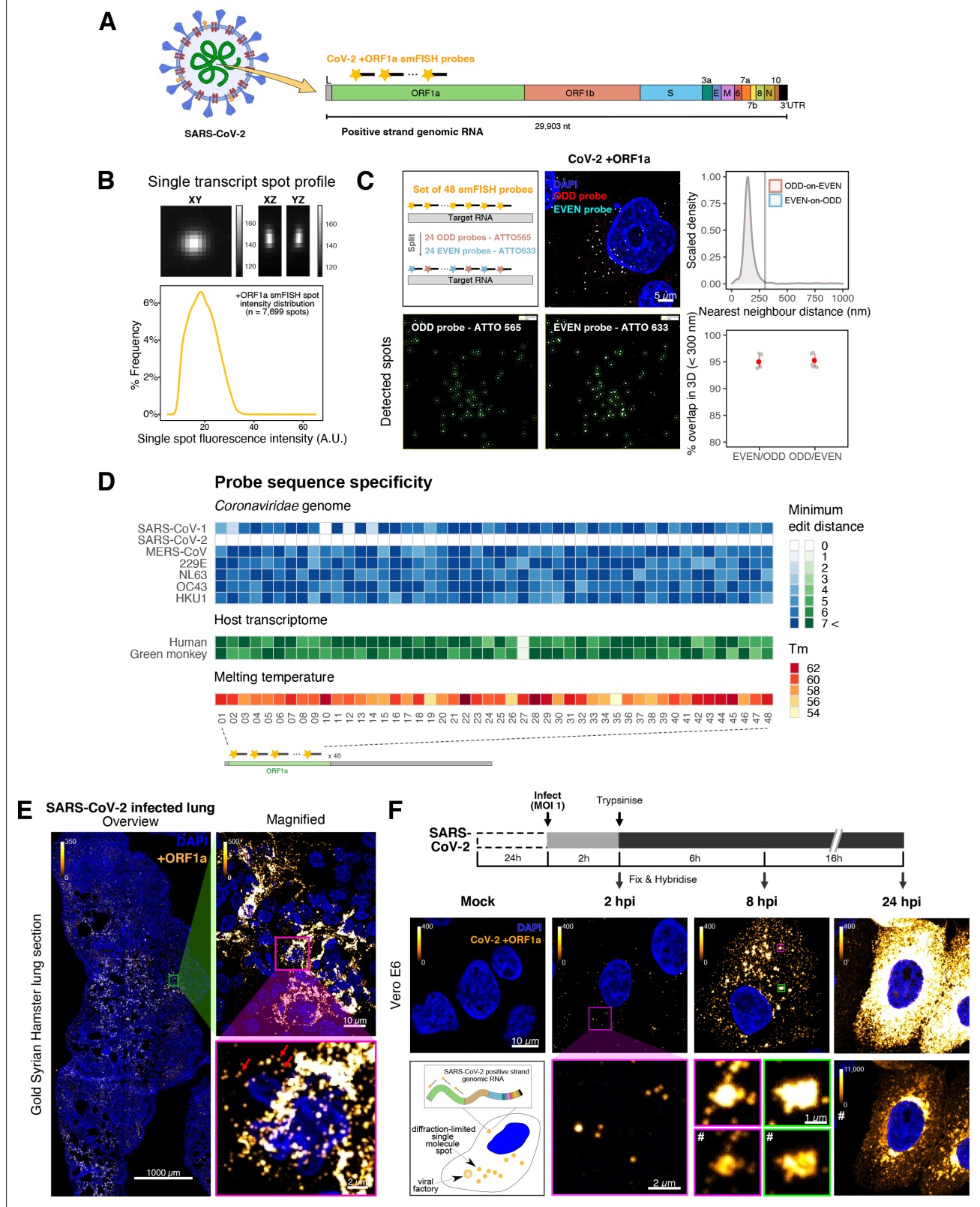

**Figure 1.** Sensitive single-molecule detection of severe acute respiratory syndrome coronavirus 2 (SARS-CoV-2) genomic RNA in infected cells. (**A**) Schematic illustration of single-molecule fluorescence in situ hybridisation (smFISH) for detecting SARS-CoV-2 positive strand genomic RNA (+gRNA) within infected cells. (**B**) Reference spatial profile of a diffraction-limited +ORF1a smFISH spot. The calibration bar represents relative fluorescence intensity (top). Frequency distribution of smFISH spot intensities, exhibiting a unimodal distribution (bottom). (**C**) Assessment of smFISH detection

*Figure 1 continued on next page*

*Figure 1 continued*

sensitivity by a dual-colour co-detection method. Maximum intensity projected images and corresponding FISH-quant spot detection views of ODD and EVEN probe sets are shown. Scale bar = 5 µm. Density histogram of nearest-neighbour distance from one spectral channel to another (top). Vertical line indicates 300 nm distance. Percentage overlap between spots detected by ODD and EVEN split probes, calculated bidirectionally (bottom). (**D**) Heatmap of probe sequence alignment against various *Coronaviridae* and host transcriptomes. Each column represents individual 20 nt + ORF1a probe sequences. The minimum edit distance represents mismatch scores, where '0' indicates a perfect match. Melting temperatures of each probe at the smFISH hybridisation condition are shown. (**E**) smFISH against +ORF1a in SARS-CoV-2-infected formalin-fixed paraffin-embedded (FFPE) lung tissue from Golden Syrian hamster at 4 days post infection. Hamsters were infected intranasally with $5 \times 10^4$ plaque-forming unit (PFU) of SARS-CoV-2 BVICO1. At necropsy, lung samples were fixed in 10% buffered formalin and embedded in paraffin wax. Red arrows in magnified panels indicate single-molecule RNA spots. Scale bars = 1000, 10, or 2 µm. (**F**) Experimental design for visualising SARS-CoV-2 gRNA with smFISH at different timepoints after infection of Vero E6 cells. Cells were seeded on cover-glass and 24 hr later inoculated with SARS-CoV-2 (Victoria [VIC] strain at multiplicity of infection [MOI] 1) for 2 hr. Non-internalised viruses were removed by trypsin digestion and cells fixed at the timepoints shown. Representative 4 µm maximum intensity projection confocal images are shown. The calibration bar labelled with the symbol '#' is used to display wider dynamic contrast range. Magnified view of insets in the upper panels is shown in lower panels. Scale bars = 10 µm or 2 µm.

The online version of this article includes the following figure supplement(s) for figure 1:

**Figure supplement 1.** Specific detection of severe acute respiratory syndrome coronavirus 2 (SARS-CoV-2) RNA using single-molecule fluorescence in situ hybridisation (smFISH).

that localised to the perinuclear region (*Figure 1F*); consistent with the reported association of viral replication factories with membranous structures derived from the endoplasmic reticulum (*V'kovski et al., 2021*). Interestingly, our observation of individual gRNA molecules at the periphery of cells (*Figure 1F*) is also consistent with individual viral particles observed at the same location by electron microscopy (*Cortese et al., 2020*; *Klein et al., 2020*). We conclude that detection of SARS-CoV-2 + gRNA by smFISH identifies changes in viral RNA abundance and cellular distribution during early replication. Our method detected +gRNA molecules in all the expected subcellular locations, namely in virions, free in the cytoplasm and in clusters at the periphery of the nucleus, reflecting different steps of the viral life cycle (*Cortese et al., 2020*; *Hackstadt et al., 2021*; *Mendonça et al., 2021*).

## Quantification of SARS-CoV-2 genomic and subgenomic RNAs

SARS-CoV-2 produces both gRNA and subgenomic (sg)RNAs which are both critical to express the full scope of viral proteins in the right time and stoichiometry. However, quantitation of sgRNAs is challenging due to their sequence overlap with the 3' end of the gRNA. To estimate the abundance of sgRNAs, we designed two additional probe sets labelled with different fluorochromes; a +ORFN set that hybridises to all canonical positive sense viral RNAs, and a +ORFS set that detects only sgRNA encoding S (S-sgRNA) and gRNA (*Figure 2A*; *Kim et al., 2020*). Therefore, spots showing fluorescence only for +ORFN or +ORFS probe sets will represent sgRNAs, whereas spots positive for both +ORFN or +ORFS and +ORF1a will correspond to gRNA molecules. We applied this approach to visualise SARS-CoV-2 RNAs in infected Vero E6 cells (6 hpi) and observed a high abundance of sgRNAs compared to gRNAs (*Figure 2B*), in agreement with RNA sequencing studies (*Alexandersen et al., 2020*; *Kim et al., 2020*). Further analysis revealed that the +ORFN single-labelled sgRNAs were more uniformly dispersed throughout the cytoplasm than dual-labelled gRNA, consistent with their predominant role as mRNAs to direct protein synthesis (*Figure 2—figure supplement 1*). However, gRNAs were enriched near the periphery of the nucleus in a clustered fashion. Association of gRNA with nucleocapsid (N) is essential for the assembly of coronavirus particles (*Carlson et al., 2020*; *Dinesh et al., 2020*; *Iserman et al., 2020*). To monitor this process in SARS-CoV-2, we combined smFISH using the +ORF1a and +ORFN probe sets with immunofluorescence detection of the viral nucleocapsid (N). Our findings show that N protein primarily co-localises with gRNA, while displaying a limited overlap with sgRNAs (*Figure 2C*, *Figure 2—figure supplement 2*). Together, these data demonstrate the specificity of our probes to accurately discriminate between the gRNA and sgRNAs.

Negative sense gRNA and sgRNAs are the templates for the synthesis of positive sense RNAs and are expected to localise to viral replication factories. However, their detection by RT-qPCR or sequencing is hampered by cDNA library protocols that employ oligo(dT) selection and by primer binding to dsRNA structures (*Ramanan et al., 2016*; *Sethna et al., 1991*). To detect negative sense viral RNAs, we denatured dsRNA complexes through either formamide, DMSO, or sodium hydroxide treatment (*Singer et al., 2021*; *Wilcox et al., 2019*). The combination of DMSO with heat treatment

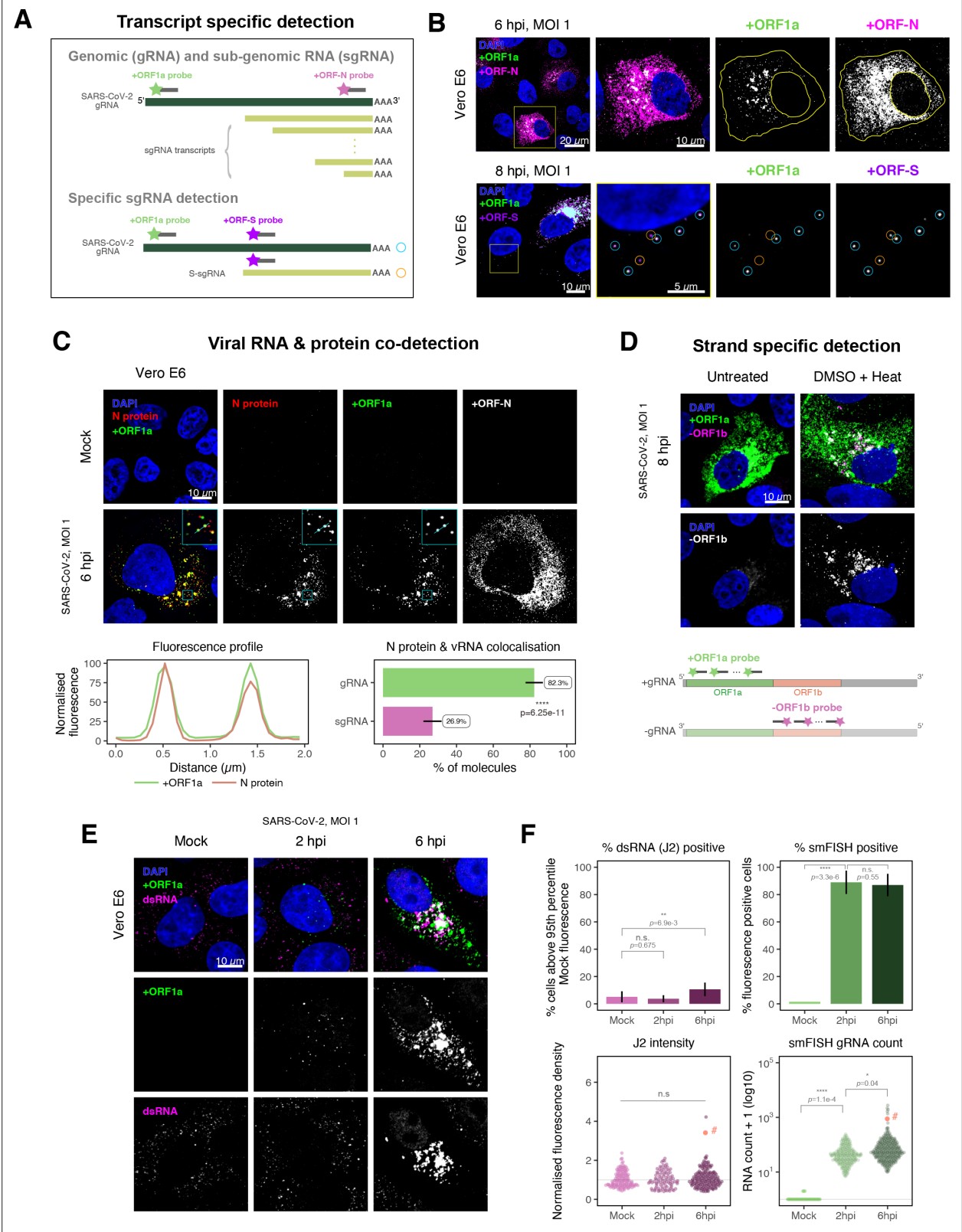

**Figure 2.** Dissecting severe acute respiratory syndrome coronavirus 2 (SARS-CoV-2) gene expression using single-molecule fluorescence in situ hybridisation (smFISH). (**A**) Schematic illustration of transcript-specific targeting of SARS-CoV-2 genomic RNA (gRNA) and subgenomic RNA (sgRNA) using smFISH. (**B**) Transcript-specific visualisation of gRNA and sgRNA in infected (Victoria [VIC] strain) Vero E6 cells. Cells were infected with SARS-CoV-2 (VIC strain) and hybridised with probes against +ORF1a and +ORFN probe at 6 hr post infection (hpi) (upper panels) or +ORF1a and +ORFS

*Figure 2 continued on next page*

*Figure 2 continued*

probe at 8 hpi (lower panels). Representative 3 µm maximum intensity projected confocal images are shown. Orange circular regions of interest (ROIs) indicate S-sgRNA encoding Spike, whereas dual-colour spots (teal-coloured ROIs) represent gRNA. Scale bar = 5, 10, or 20 µm. (C) Co-detection of viral nucleocapsid (N) with gRNA and sgRNA. Monoclonal anti-N (Ey2A clone) was used for N protein immunofluorescence. Representative 3 µm z-projected confocal images are shown. The inset shows a magnified view of co-localised N and gRNA. Scale bar = 10 µm. Fluorescence profiles of N immunostaining and gRNA smFISH intensity across a 2 µm linear distance are shown in the image inset (lower left). Percentage of co-localised gRNA or sgRNA molecules with N protein at 6 hpi. Co-localisation was assessed by N fluorescence density within point-spread function ellipsoids of RNA spots over random coordinates. sgRNA were defined as single-coloured spots with +ORFN probe signal only (n = 7) (lower right). Student's *t*-test. ****p<0.0001. (D) Detection of both positive and negative genomic RNA by denaturing viral double-stranded RNA (dsRNA) with DMSO and heat treatment at 80°C (upper panels). 3 µm z-projected images of infected Vero E6 cells at 8 hpi are shown. Scale bar = 10 µm. Schematic illustration of +ORF1a and -ORF1b probe targeting regions (lower panel). -ORF1b probe target region does not overlap with +ORF1a target sequences to prevent probe duplex formation. (E) Comparison of anti-dsRNA (J2) and gRNA smFISH. Full z-projected images of infected Vero E6 cells co-stained with J2 and smFISH are shown. Scale bar = 10 µm. (F) Percentage of infected cells detected by J2 or smFISH (upper panels). For J2-based quantification, we defined the threshold as 95th percentile fluorescent signal of uninfected cells (Mock) due to the presence of endogenous host-derived signals. Fluorescent positive signals were used for smFISH-based quantification. Data are presented as mean ± SD. Comparison of quantification results between J2 stain and smFISH (lower panels). Each symbol represents one cell. J2 signal was quantified by fluorescence density over 3D cell volume, which was normalised to the average signal of uninfected control cells (horizontal dotted line). gRNA count represents sum of single-molecule spots and decomposed spots within viral factories. The symbol denoted with '#' is the infected cell shown in *Figure 2E* (J2 stain, n = 3 independent repeats; smFISH, n = 4). One-way ANOVA and Tukey post-hoc test. n.s., not significant; *p<0.05; **p<0.01; ****p<0.0001.

The online version of this article includes the following figure supplement(s) for figure 2:

**Figure supplement 1.** Subcellular RNA dispersion of severe acute respiratory syndrome coronavirus 2 (SARS-CoV-2) genomic RNA (gRNA) and subgenomic RNAs (sgRNAs).

**Figure supplement 2.** Preferential co-localisation of severe acute respiratory syndrome coronavirus 2 (SARS-CoV-2) nucleocapsid protein with genomic RNA (gRNA).

**Figure supplement 3.** Denaturation of severe acute respiratory syndrome coronavirus 2 (SARS-CoV-2) double-stranded RNA (dsRNA) for negative strand detection.

resulted in a loss of anti-dsRNA J2 signal, while maintaining cell integrity, suggesting a disruption of dsRNA hybrids (*Figure 2—figure supplement 3A*). We designed an smFISH probe set specific for the ORF1b antisense sequence that targets the negative sense gRNA (-gRNA) and resulted in intense diffraction-limited spots in DMSO and heat-treated cells (*Figure 2D*). The -gRNA spots were detected at a significantly lower level than their +gRNA counterparts, with substantial overlap observed between the two strands at multi-RNA spots, consistent with these foci representing active sites of viral replication. To determine if these multi-RNA foci contain dsRNA, the permeabilised infected cells were treated with RNaseT1 or RNaseIII, which are nucleases specific for single-stranded RNA (ssRNA) and dsRNA, respectively (*Figure 2—figure supplement 3B*). RNaseT1 digestion diminished the +ORF1a probe signal, while RNaseIII treatment abolished the anti-dsRNA J2 signal. A cocktail of RNaseT1 and RNaseIII ablated both +ORF1a probe binding and anti-dsRNA J2 signals, demonstrating that the +ORF1a probe set hybridises to both single and duplex RNA under our experimental conditions. Furthermore, treating cells with DMSO prior to RNaseT1 fully ablated the smFISH signal (*Figure 2—figure supplement 3C*), demonstrating that denaturation makes dsRNA accessible for RNaseT1 degradation. In summary, our data show that probe binding to negative strand gRNA requires chemical denaturation, suggesting that this replication intermediate is rich in dsRNA structures.

## Anti-dsRNA antibodies underestimate SARS-CoV-2 replication

The establishment of replication factories is a critical phase of the virus life cycle. Previous reports have identified these viral factories using the J2 dsRNA antibody (*Burgess and Mohr, 2015*; *Cortese et al., 2020*; *Targett-Adams et al., 2008*; *Weber et al., 2006*). However, this approach depends on high levels of viral dsRNA as cells naturally express endogenous low levels of dsRNA (*Dhir et al., 2018*; *Kimura et al., 2018*; *Figure 2E*). To evaluate the ability of J2 antibody to quantify SARS-CoV-2 replication sites, we co-stained infected cells at 2 and 6 hpi with both J2 and +ORF1a smFISH probes. No viral-specific J2 signal was detected at 2 hpi, and only 10% of infected cells stained positive at 6 hpi, in agreement with previous observations (*Cortese et al., 2020*; *Eymieux et al., 2021*; *Figure 2E*). In contrast, more than 85% of the cells showed diffraction-limited smFISH signals at both timepoints (*Figure 2F*). Furthermore, the average J2 signal detected in the SARS-CoV-2-infected cells at both timepoints was comparable to uninfected cells (*Figure 2F*). These data clearly show that

the J2 antibody, although broadly used, underestimates the frequency of SARS-CoV-2 infection. In contrast, smFISH detected gRNA as early as 2 hpi, with a significant increase in copy number by 6 hpi, highlighting its utility to detect and quantify viral replication factories.

## SARS-CoV-2 replication at single-molecule resolution

The efficiency and sensitivity of smFISH to detect single molecules of SARS-CoV-2 RNA allowed us to investigate the dynamics of viral replication in Vero E6 cells during the first 10 hr of infection (*Figure 3A*). At 2 hpi, the +ORF1a probe set detected predominantly single molecules of +gRNA with a median value of ~30 molecules per cell (*Figure 3B and C*). Interestingly, at 2 hpi RDV treatment did not affect the number of gRNA copies per cell, suggesting that these RNAs derive from incoming viral particles (*Figure 3C*). In contrast, the increase in gRNA copies per cell at 4 and 6 hpi was inhibited by RDV, indicating active viral replication. The infected cell population showed varying gRNA levels that we classified into three groups; (i) 'partially resistant' cells with $<10^2$ gRNA copies that showed no increase in gRNA burden between 2 and 8 hpi (60% of the population); (ii) 'permissive' cells with $~10^2$–$10^5$ copies per cell showing a modest increase over time (~30%); and (iii) 'super-permissive' cells with $>10^5$ copies per cell showing a sharp increase in gRNA copies (~10%). Given the high gRNA density in super-permissive cells, RNA counts were estimated by correlating the integrated fluorescence intensity of reference single molecules to the total fluorescence of the 3D cell volume (see Materials and methods), which follows a linear relationship (*Figure 3—figure supplement 1*). Analysing the total cellular gRNA content showed that 'super-permissive' cells are the dominant source of gRNA across the culture (*Figure 3D*). This suggests that bulk RNA analyses such as RT-qPCR are biased towards this high gRNA burden group. Importantly, we found that cellular heterogeneity persists beyond the initial hours of infection. Even at 24 hpi, 40% of the cells did not reach the super-permissive state, and they formed a distinct population with approximately 10-fold less gRNA (*Figure 3C and E*). Similar heterogeneous cell populations were observed between 24 and 48 hpi, although the overall levels of gRNA started to decline after 32 hpi, reflecting cytotoxic effects and virus egress (*Figure 3—figure supplement 2A–C*). Therefore, these results highlight a wide variation in cell susceptibility to SARS-CoV-2 replication, which persisted throughout the infection. Notably, the high level of gRNA content in super-permissive cells (~$10^7$ counts/cell) was similar throughout the time course, suggesting the existence of an upper limit of gRNA copies in Vero E6 cells (*Figure 3C*).

Viral RNA stability is an important determinant for the virus' ability to initiate and maintain a productive infection. In an effort to determine the stability of SARS-CoV-2 RNA, we added RDV simultaneously to SARS-CoV-2 infection and followed gRNA persistence in a time course. Notably, the average number of gRNA copies per cell was stable in RDV-treated cells (*Figure 3C*), suggesting that the incoming gRNA is long-lived. To assess if gRNA is also stable at later times post infection, we treated cells with RDV at 24 hpi and measured gRNA abundance at different times post treatment (*Figure 3—figure supplement 2A–C*). As expected, RDV treatment led to a reduced proportion of super-permissive cells and non-viable cells at 48 hpi, indicating an inhibition of viral replication and, consequently, viral-induced cell death (*Figure 3—figure supplement 2D*). However, considerable levels of gRNA persisted in these cells even after 24 hr of RDV treatment, suggesting that gRNA is also relatively stable at late times post infection. To estimate the half-life of gRNA in late infection, we fitted a decay curve and calculated the half-life of gRNA within a range of 6–8 hr (*Figure 3—figure supplement 2C*). This half-life might be underestimated as gRNA loss is not only due to decay but also to virus egress. Moreover, while we used an RDV dose that exceeded the 90% inhibitory concentration ($IC_{90}$) (*Figure 3—figure supplement 3*), we cannot rule out that incomplete inhibition by RDV could affect our half-life estimates (*Figure 3C*).

Simultaneous analysis of +ORF1a and +ORFN revealed similar expression kinetics for sgRNA, with 11 copies/cell of sgRNA detected in 63% of infected cells at 2 hpi (*Figure 3C and F*). Since +sgRNA requires -sgRNA template for its production, our results imply that multiple rounds of transcription occur rapidly following virus internalisation that are RDV insensitive. By 6 hpi, most cells contain sgRNA (*Figure 3F*), with the super-permissive cells supporting high levels of sgRNA transcription. We examined the vRNA replication dynamics and found the ratio of sgRNA/gRNA ranged from 0.5 to 8 over time (*Figure 3G*), consistent with a recent report in diagnostic samples (*Alexandersen et al., 2020*). Notably, the sgRNA/gRNA ratio increased between 2 and 10 hpi, followed by a decline at 24 hpi, indicating a shift in preference to produce gRNA over sgRNA in later stages of infection. A

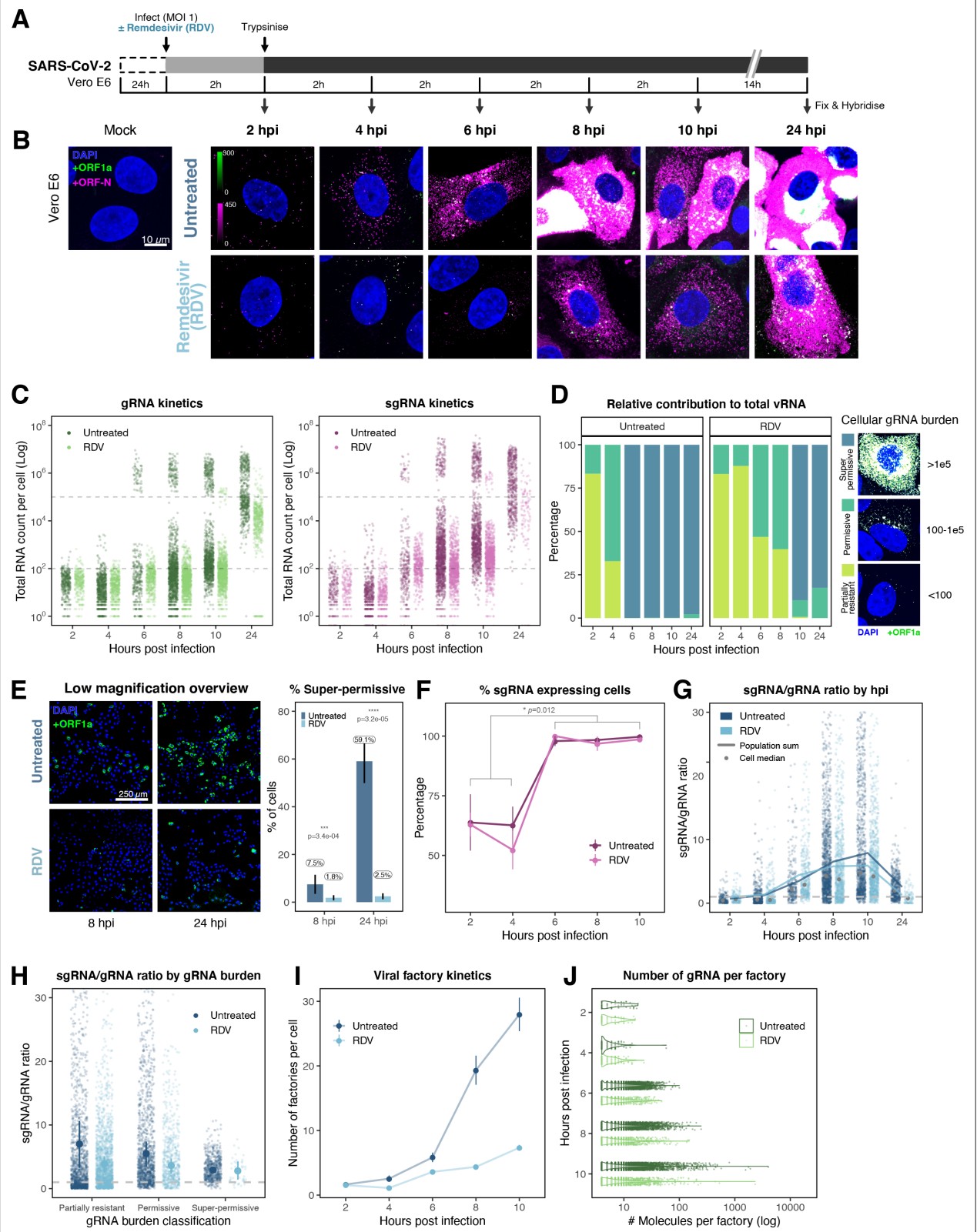

**Figure 3.** Profiling severe acute respiratory syndrome coronavirus 2 (SARS-CoV-2) replication kinetics at single-molecule resolution. (**A**) Experimental design to profile SARS-CoV-2 replication kinetics using single-molecule fluorescence in situ hybridisation (smFISH). Vero E6 cells were seeded on cover-glass and 24 hr later inoculated with SARS-CoV-2 (Victoria [VIC] strain, multiplicity of infection [MOI] = 1) for 2 hr. Non-internalised viruses were removed by trypsin digestion and cells fixed at the timepoints shown for hybridisation with +ORF1a and +ORFN probes. In the remdesivir (RDV) condition,

*Figure 3 continued on next page*

*Figure 3 continued*

the drug was added to cells at 10 µM during virus inoculation and maintained for the infection period. (**B**) Maximum z-projected confocal images of infected cells. Intensity calibration bars are shown for the +ORF1a and +ORFN channels. Scale bar = 10 µm. (**C**) Bigfish quantification of +gRNA or +sgRNA counts per cell. sgRNA counts were calculated by subtracting +ORF1a counts from +ORFN counts per cell. Horizontal dashed lines indicate $10^2$ or $10^5$ molecules of RNA. 24 hr post infection (hpi) samples and the cells harbouring >$10^7$ RNA counts were quantified by extrapolating single-molecule intensity. Quantified cells from all replicates are plotted (2–8 hpi, n ≥ 3; 10 and 24 hpi, n = 2). Number of cells analysed (untreated/RDV): 2 hpi, 373/273; 4 hpi, 798/516; 6 hpi, 370/487; 8 hpi, 1442/1022; 10 hpi, 1175/1102; 24 hpi, 542/249. (**D**) Relative contribution of viral gRNA within the infected cell population. The infected cells were classified into three groups based on gRNA counts: (i) 'partially resistant' – gRNA <100; 'permissive' – 100 < gRNA < $10^5$; 'super-permissive' – gRNA >$10^5$. The total gRNA within the infected wells was obtained by summing gRNA counts in population, and the figure shows the relative fraction from each classification. Representative max-projected images of cells in each category are shown (2–8 hpi, n ≥ 3; 10 and 24 hpi, n = 2). (**E**) Identification of super-permissive cells from a low-magnification (×20) high-throughput smFISH assay (left panels). Full z-projected (9 µm) images of infected Vero E6 cells hybridised with +ORF1a probes are shown. At this magnification, only the cells with vRNA count >~$10^5$ (i.e. corresponding to 'super-permissive' cells) exhibit significant smFISH fluorescence compared to uninfected cells. Scale bar = 250 µm. Percentage of super-permissive cells in untreated and RDV-treated conditions at 8 and 24 hpi (right panel). Labels represent average values. Data are represented as mean ± SD (n = 3, ~ 2000 cells were scanned from each replicate well). Student's *t*-test. ***p<0.001; ****p<0.0001. (**F**) Percentage of infected cells expressing sgRNA. sgRNA-expressing cells were identified by those having a (ORF-N – ORF1a) probe count more than 1. Data are represented as mean ± SEM (2–8 hpi, n ≥ 3; 10 and 24 hpi, n = 2). (**G**) Per cell ratio of sgRNA/gRNA counts across the time series. Grey symbols represent cell-to-cell median values, whereas the line plot represents ratio calculated from population sum of gRNA and sgRNA. The number of cells analysed is the same as in (*Figure 3C*), with the exception of cells having equal ORF1a and ORF-N probe counts. Horizontal dashed line represents value of 1 (2–8 hpi, n ≥ 3; 10 and 24 hpi, n = 2). (**H**) Per cell ratio of sgRNA/gRNA counts grouped by gRNA burden classification as defined in *Figure 3D*. Data are represented as median ± SEM. Horizontal dashed line represents value of 1 (2–8 hpi, n ≥ 3; 10 and 24 hpi, n = 2). (**I**) The number of viral factories per cell increase over time as assessed by smFISH cluster detection. Cells harbouring >$10^7$ copies of RNA, less than 10 molecules of RNA, cells with no viral factories, and cells from 24 hpi timepoints were excluded from this analysis. Data are represented as mean ± SEM. Number of cells analysed (untreated/RDV): 2 hpi, 494/240; 4 hpi, 758/494; 6 hpi, 315/417; 8 hpi, 933/877; 10 hpi, 726/885 (2–8 hpi, n ≥ 3; 10 and 24 hpi, n = 2). (**J**) The kinetics of gRNA copies within viral factories. Spatially extended viral factories were resolved by cluster decomposition to obtain single-molecule counts. The type and number of cells analysed are the same as in *Figure 3I* (2–8 hpi, n ≥ 3; 10 and 24 hpi, n = 2). gRNA, genomic RNA; sgRNA, subgenomic RNA.

The online version of this article includes the following figure supplement(s) for figure 3:

**Figure supplement 1.** Correlation between single-molecule fluorescence in situ hybridisation (smFISH) RNA counts and smFISH fluorescence intensity.

**Figure supplement 2.** The dynamics and heterogeneity of severe acute respiratory syndrome coronavirus 2 (SARS-CoV-2) RNA replication.

**Figure supplement 3.** Remdemsivir (RDV) dose-response of severe acute respiratory syndrome coronavirus 2 (SARS-CoV-2) RNA replication.

similar trend was observed in RDV-treated cells, with a reduced sgRNA/gRNA peak at 8–10 hpi. We estimated the sgRNA/gRNA ratio for individual cells and found that sgRNA synthesis is favoured in the 'partially resistant' and 'permissive' cells, whereas the 'super-permissive' cells had a reduced ratio of sgRNA/gRNA (*Figure 3H*). In summary, these results indicate that gRNA synthesis is favoured in the late phase of infection, which may reflect the requirement of gRNA to assemble new viral particles.

Positive sense RNA viruses, including coronaviruses, utilise host membranes to generate viral factories, which are sites of active replication and/or virus assembly (*Wolff et al., 2020*). Our current knowledge on the genesis and dynamics of these factories in SARS-CoV-2 infection is limited. We exploited the spatial resolution of smFISH to study these structures, which we define as spatially extended foci containing multiple gRNA molecule clusters. These clusters are compatible in size with the double membrane vesicles (DMVs) employed by SARS-CoV-2 to replicate and assemble new virions, as previously identified by EM (see Materials and methods; *Cortese et al., 2020*; *Mendonça et al., 2021*). We refer to these gRNA clusters as 'factories'. We observed 1–2 factories per cell at 2 hpi, which increased to ~30 factories/cell by 10 hpi (*Figure 3I*). In addition, the average number of gRNA molecules within these factories, although variable, increased over time (*Figure 3J*). RDV treatment reduced both the number of viral factories per cell and their RNA content. Together these data show the capability of smFISH to localise and quantify active sites of SARS-CoV-2 replication and to measure changes in gRNA and sgRNA at a single-cell level over the course of the infection.

## Super-permissive cells are randomly distributed

Our earlier kinetic analysis of infected Vero E6 cells identified a minor population of 'super-permissive' cells containing high gRNA copies at 8 hpi. A random selection of ~300 cells allowed us to further characterise the infected cell population (*Figure 4A and B*). To extend these observations, we examined the vRNAs in two human lung epithelial cell lines, A549-ACE2 and Calu-3, that are widely used to study SARS-CoV-2 infection (*Chu et al., 2020*; *Hoffmann et al., 2020*). In agreement with our earlier

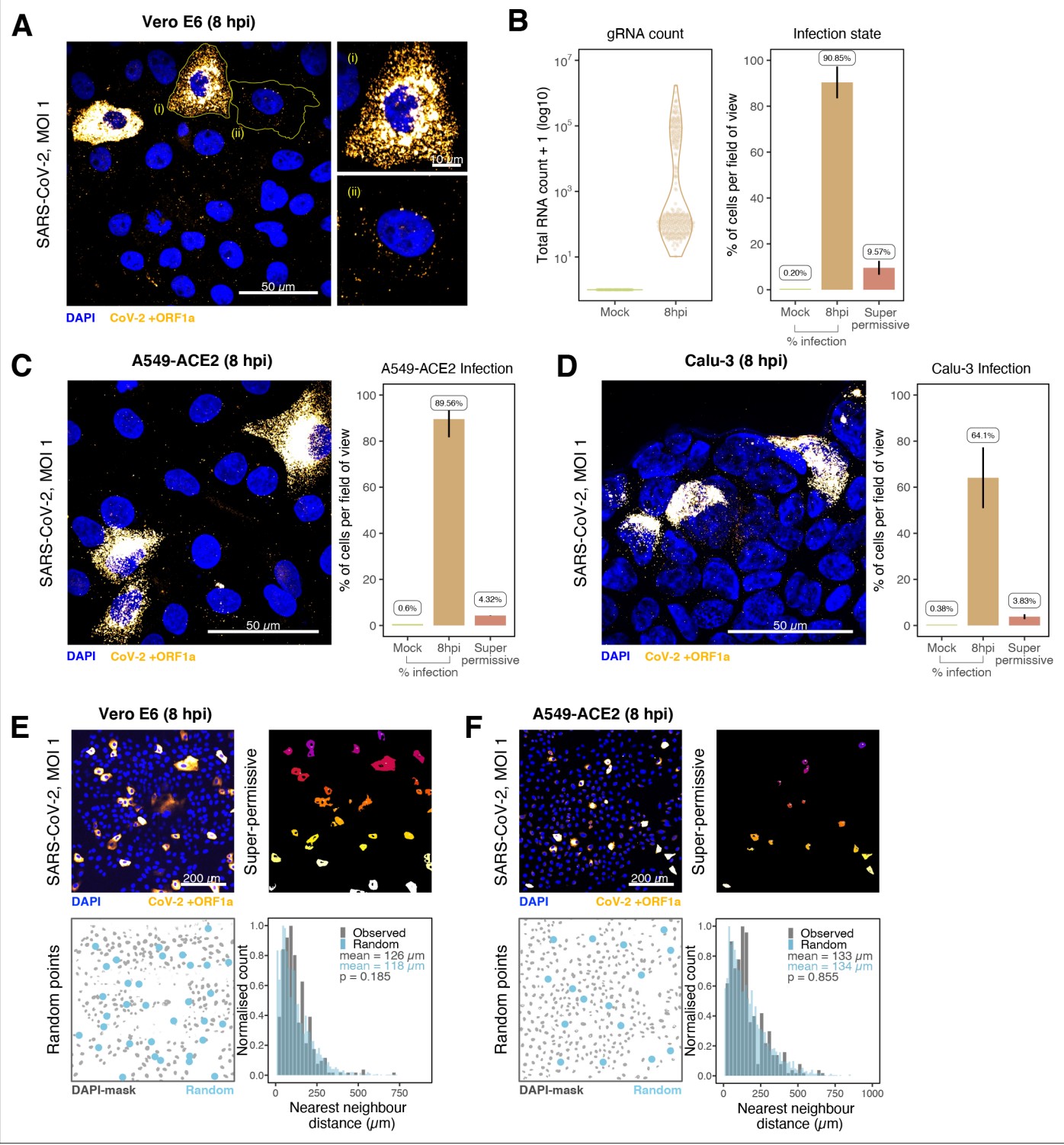

**Figure 4.** Heterogeneous severe acute respiratory syndrome coronavirus 2 (SARS-CoV-2) RNA replication. (**A**) Representative ×60 magnified field of view (FOV) of SARS-CoV-2-infected Vero E6 cells at 8 hr post infection (hpi) (Victoria [VIC] strain, multiplicity of infection [MOI] = 1). Single-molecule fluorescence in situ hybridisation (smFISH) against ORF1a was used to visualise cellular heterogeneity in viral RNA counts. Magnified panels show (i) a 'super-permissive' cell and (ii) a cell with discrete viral RNA copies. Scale bar = 10 or 50 µm. (**B**) Discrete separation of genomic RNA (gRNA) count distribution among infected cells randomly sampled at 8 hpi, where each symbol represents a cell. Statistics for the percentage of infected cells and frequency of 'super-permissive' cells at 8 hpi. Quantification was performed per FOV, and the number labels represent average values. Cells with >10⁵ gRNA copies were considered to be 'super-permissive' (gRNA quantification: n = 4, 148 uninfected and 316 infected cells; percentage

*Figure 4 continued on next page*

*Figure 4 continued*

infection: n = 3). (**C, D**) Heterogeneous SARS-CoV-2 replication in lung epithelial A549-ACE2 and Calu-3 cells. The percentage of infected and super-permissive cells was quantified as with Vero E6 cells above. Scale bar = 50 µm (A549-ACE2, n = 2; Calu-3, n = 3). (**E, F**) Spatial distribution analysis of super-permissive Vero E6 and A549-ACE2 cells at 8 hpi. Low-magnification smFISH overview of infected cells (top left). 2D mask of super-permissive cells (top right). An example of randomly simulated points within the DAPI mask (bottom left). Same number of random points as super-permissive cells were simulated 10 times per FOV. Histogram of nearest-neighbour distances calculated from super-permissive cells (Observed) and randomly simulated points (Random) (bottom right). Further modes of spatial analyses are presented in *Figure 4—figure supplement 1* with infected Calu-3 cells. All confocal images are presented as maximum full z-projection. Data are represented as mean ± SD. Student's *t*-test; p-values are shown on the presented visual (Vero E6, n = 3; A549-ACE2, n = 2).

The online version of this article includes the following figure supplement(s) for figure 4:

**Figure supplement 1.** Spatial distribution of super-permissive severe acute respiratory syndrome coronavirus 2 (SARS-CoV-2)-infected cells.

observations with Vero E6, 3–5% of A549-ACE2 and Calu-3 cells showed a 'super-permissive' phenotype (*Figure 4C and D*). An important question is how these 'super-permissive' cells are distributed in the population as the pattern could highlight potential drivers for susceptibility (*Healy et al., 2020*). Infection can induce innate signalling that can lead to the expression and secretion of soluble factors such as interferons that induce an antiviral state in the local cellular environment (*Belkowski and Sen, 1987*; *Schoggins and Rice, 2011*). Regulation can be widespread through paracrine signalling or affect only proximal cells. We considered three scenarios where 'super-permissive' cells are randomly distributed, evenly separated or clustered together. We compared the average nearest-neighbour distance between 'super-permissive' cells and simulated points that were distributed either randomly, evenly, or in clusters (*Figure 4—figure supplement 1*). In summary, our results show conclusively that the 'super-permissive' infected Vero E6, A549-ACE2, and Calu-3 cells were randomly distributed (*Figure 4E and F*, *Figure 4—figure supplement 1*). We interpret these data as being consistent with an intrinsic property of the cell that defines susceptibility to virus infection. The data also argue against cell-to-cell signalling mechanisms that would either lead to clustering (if increasing susceptibility) or to an even distribution (if inhibiting) of infected cells.

## Differential replication kinetics of the B.1.1.7 and VIC strains

The recent emergence of SARS-CoV-2 VOCs, which display differential transmission, pathogenesis, and infectivity, has changed the course of the COVID-19 pandemic. Recent studies have focused on mutations in the Spike protein and whether these alter particle uptake into cells and resistance to vaccine or naturally acquired antibodies (*Collier et al., 2021*; *Dicken et al., 2021*; *Planas et al., 2021*). The B.1.1.7 variant is associated with higher transmission (*Davies et al., 2021*; *Galloway et al., 2021*; *Volz et al., 2021*) and has 17 coding changes mapping to both non-structural (ORF1a/b, ORF3a, ORF8) and structural (Spike and N) proteins. Mutations within the non-structural genes could affect virus replication, independent of Spike-mediated entry, thus we used smFISH to compare the replication kinetics of the B.1.1.7 and VIC strains (*Figure 5A*). We discovered that the number of gRNA molecules at 2 hpi was similar for both viruses, reflecting similar cell uptake of viral particles (*Figure 5B–E*). However, the quantities of intracellular gRNA and sgRNA were lower in B.1.1.7-infected cells compared to VIC at 6 and 8 hpi (*Figure 5E*). We also found that while the amount of gRNA per cell was reduced in the B.1.1.7 variant, there were an equal number of +ORF1a and +ORFN-positive cells (*Figure 5D*), suggesting that the reduced B.1.1.7 RNA burden is due to a differential replication efficiency rather than infection rate. The B.1.1.7 variant also showed a reduced number of replication factories per cell (*Figure 5F*), with each focus containing on average a lower number of gRNA molecules compared to the VIC strain (*Figure 5G*). RDV treatment ablated the differences between the viral strains, demonstrating that the observed phenotype is replication-dependent (*Figure 5B,E-I*). Nevertheless, the lower level of individual gRNA that we detected in RDV-treated cells persisted for at least 8 hpi in both the VIC and B.1.1.7 strains. We conclude that individual gRNA molecules of both the strains are highly stable in the cytoplasm of infected cells.

Consistent with the delay in replication, we observed a shallower growth of the sgRNA/gRNA ratio in B.1.1.7-infected cells between 2 and 8 hpi compared to the VIC strain (*Figure 5H*). These differences between the strains were apparent in all three classifications of cells from our earlier gRNA burden criteria. We noted that B.1.1.7-infected 'partially resistant' and 'permissive' cells show lower sgRNA/gRNA ratio while 'super-permissive' cells displayed 1.5-fold higher ratio compared to VIC

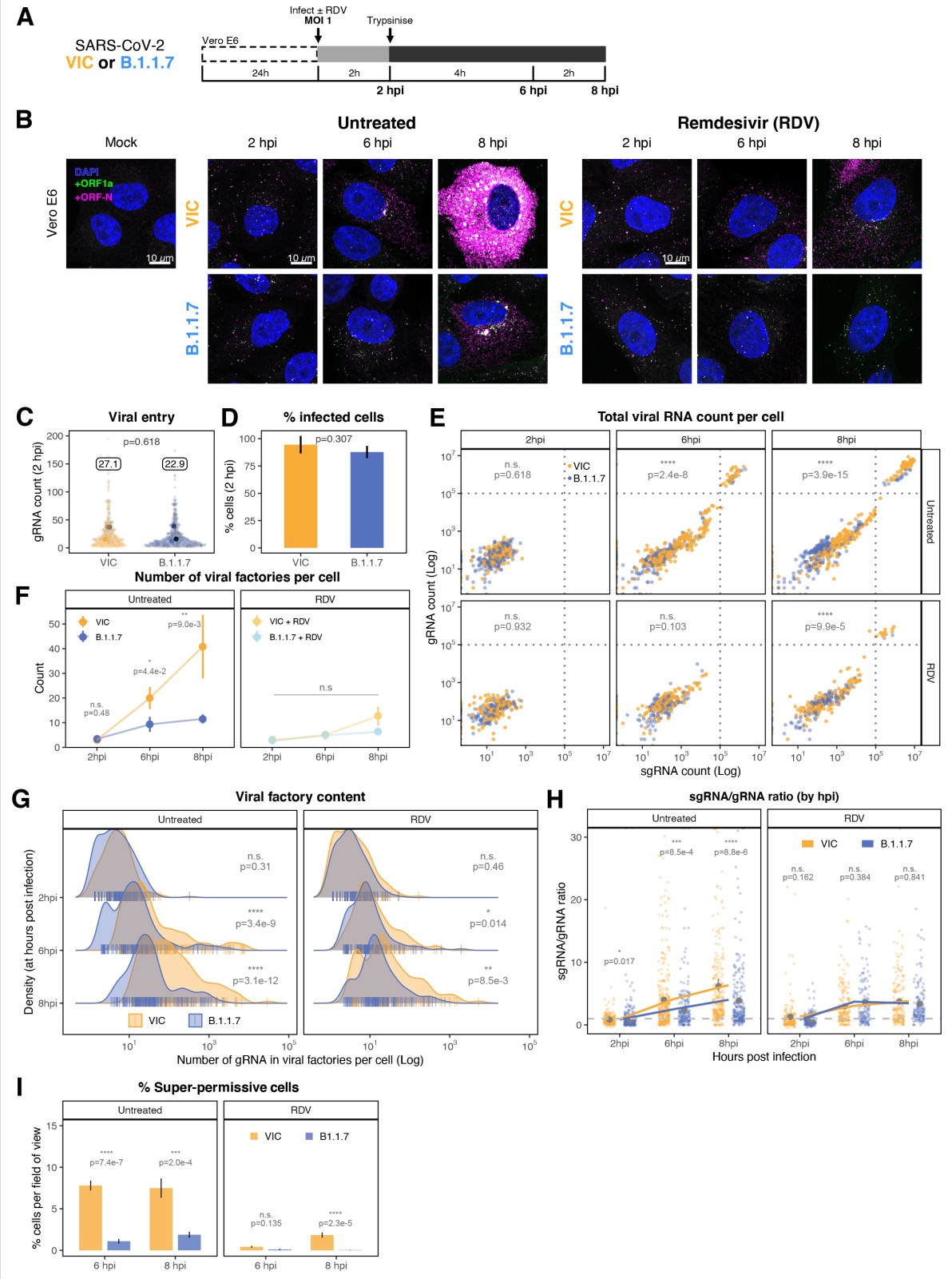

**Figure 5.** Delayed replication kinetics of B.1.1.7 variant. (**A**) Experimental design to compare the replication kinetics of Victoria (VIC) and B.1.1.7 severe acute respiratory syndrome coronavirus 2 (SARS-CoV-2) strains. Vero E6 cells were seeded on cover-glass and 24 hr later inoculated with VIC or B.1.1.7 strain (multiplicity of infection [MOI] = 1) for 2 hr. Non-internalised viruses were removed by trypsin digestion, and cells were fixed at designated timepoints for hybridisation with +ORF1a and +ORFN probes. In remdesivir (RDV) condition, the drug was added to cells at 10 μM during virus

*Figure 5 continued on next page*

*Figure 5 continued*

inoculation and maintained for the infection period. (**B**) Maximum z-projected confocal images of Vero E6 cells infected with VIC or B.1.1.7 strains. Representative super-permissive cells from the time series are shown. Scale bar = 10 µm. (**C**) Comparing viral genomic RNA (gRNA) counts at 2 hr post infection (hpi) between VIC and B.1.1.7. Each symbol represents a cell. Different hue of colours represents readings taken from individual repeat experiments, and the labels represent average values (n = 3; VIC, 424 cells; B.1.1.7, 519 cells). Mann–Whitney *U* test. (**D**) Comparing percentage of infected cells between the two viral strains at 2 hpi. Infected cells were determined by +ORF1a single-molecule fluorescence in situ hybridisation (smFISH) fluorescence. Data are represented as mean ± SD (n = 3). Student's t-test. (**E**) Bigfish quantification of gRNA and subgenomic RNA (sgRNA) smFISH counts per cell. Quantification was performed as in (*Figure 3C*). Due to bimodality of the data, statistical significance was determined using two-sample Kolmogorov-Smirnov test to compare cumulative distribution of+ ORF1 a counts between the two strains. (n = 3). (VIC, 2 : 6 : 8 hpi = 460 : 343 : 407 cells; B.1.1.7, 2 : 6 : 8 hpi = 396 : 487 : 429 cells). (**F**) Comparing the number of viral factories per cell between the two viral strains across the time series. Cells harbouring >$10^7$ copies of vRNA were excluded from analysis. Viral factories were identified using Bigfish cluster detection as with (*Figure 3I*). Data are represented as mean ± SEM (n = 3). Mann–Whitney *U* test. (**G**) Density ridge plot showing the number of gRNA copies within viral factories for VIC and B.1.1.7 variants. The density distribution represents the number of molecules per viral factories per cell. Vertical segment symbol represents a cell (n = 3). Mann–Whitney *U* test. (**H**) Per cell ratio of sgRNA/gRNA counts across the time series. Grey symbols represent cell-to-cell mean ± SE which are connected by line plots. Horizontal dashed line represents value of 1 (n = 3). Mann–Whitney *U* test. (**I**) Comparison of the percentage of super-permissive cells between the two strains assessed from low-magnification high-throughput smFISH assay (see *Figure 5—figure supplement 1* for details). Data are represented as mean ± SD (n = 3). Student's *t*-test. n.s., not significant; *p<0.05; **p<0.01; ***p<0.001; ***p<0.0001.

The online version of this article includes the following figure supplement(s) for figure 5:

**Figure supplement 1.** Delayed replication kinetics of B.1.1.7 severe acute respiratory syndrome coronavirus 2 (SARS-CoV-2) variant.

**Figure supplement 2.** Spatial distribution of Victoria (VIC) or B.1.1.7 severe acute respiratory syndrome coronavirus 2 (SARS-CoV-2) super-permissive cells.

(*Figure 5—figure supplement 1A*). The frequency of super-permissive cells was lower for B.1.1.7 at 6 and 8 hpi (*Figure 5I*, *Figure 5—figure supplement 1B and C*). In agreement with our results with VIC (*Figure 4E and F*), the distribution of super-permissive cells with B.1.1.7 was random at 8 hpi; however, this changed to a non-random pattern at 24 hpi. In contrast, the distribution of VIC super-permissive cells remained random at all timepoints (*Figure 5—figure supplement 2*). We interpret these results as demonstrating differences in the infection kinetics of the variants, with B.1.1.7 displaying a potentially higher capacity to spread locally between adjacent cells than VIC.

To test whether our findings using B.1.1.7 are applicable to other cell types, we assessed the replication of both variants in A549-ACE2 cells that were recently reported to be immunocompetent (*Li et al., 2021*). Both VIC and B.1.1.7 infections resulted in comparable numbers of infected cells and similar numbers of gRNA molecules per cell at 2 hpi, demonstrating a similar degree of viral particle entry into cells (*Figure 6—figure supplement 1A*). However, infection with the B.1.1.7 variant led to a reduced gRNA and sgRNA burden at 8 and 24 hpi (*Figure 6A and B*, *Figure 6—figure supplement 1B and C*). Moreover, fewer 'super-permissive' cells were detected at these timepoints (*Figure 6C*). To evaluate whether the slower replication kinetics of B.1.1.7 was attributable to a reduction in the secretion of new particles, we measured the level of infectious virus (*Figure 6—figure supplement 1D*). We found a modest but significant reduction in the infectious titre of B.1.1.7 compared to VIC at 8 and 24 hpi, consistent with the reduced cellular RNA burden of B.1.1.7. Considering these results together, we conclude that the replication and secretion rates of B.1.1.7 are slower than VIC in contrast to its more rapid spread in the human population.

To evaluate our observation on B.1.1.7 replication kinetics with an independent method, we sequenced ribo-depleted total RNA libraries of A549-ACE2 cells infected with B.1.1.7 or VIC for 2, 8, and 24 hr (*Figure 6A*, *Figure 6—figure supplement 2A*). As expected, the number of reads mapping to SARS-CoV-2 genome increased over time, reflecting active replication and transcription (*Figure 6D*). Reads mapping to the 3′ end of the genome increased relative to the 5′ end, reflecting the synthesis of sgRNAs. In agreement with our smFISH analysis, we detected similar levels of vRNA at 2 hpi within B.1.1.7 or VIC-infected cells, consistent with similar internalisation rates in A549-ACE2 cells (*Figure 6E*). However, the abundance of vRNAs in B.1.1.7-infected cells at 8 and 24 hpi was notably lower than with VIC-infected cells (*Figure 6E*). Furthermore, the level of B.1.1.7 RNA was almost unaltered between 2 and 8 hpi, and then increased dramatically at 24 hpi (*Figure 6E*, *Figure 6—figure supplement 2B*), contrasting with VIC-infected cells, which showed a continuous increase in vRNA over time. Together, these RNA sequencing data confirm that the B.1.1.7 variant exhibits delayed replication kinetics complementing our smFISH results.

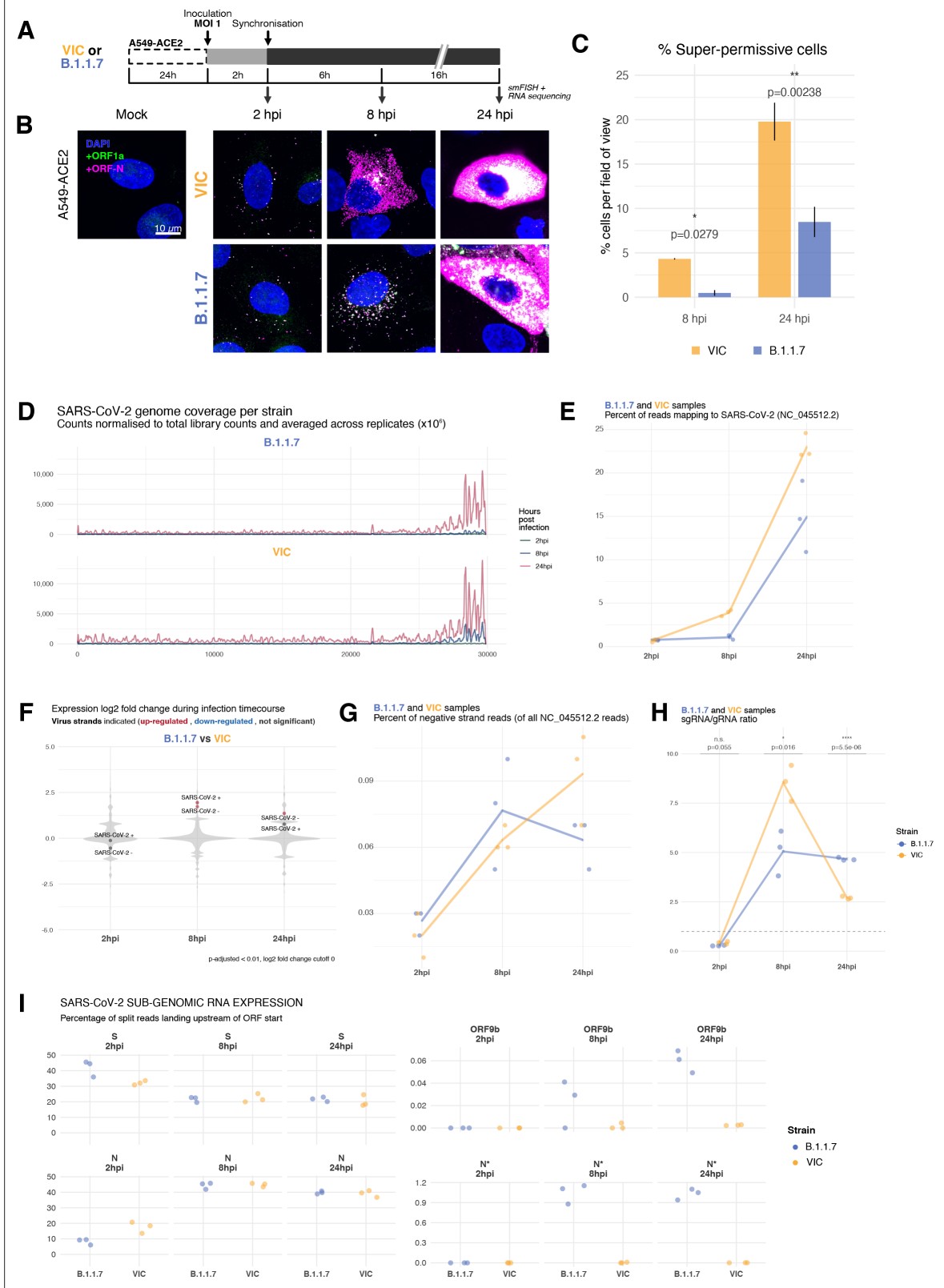

**Figure 6.** Transcriptomic landscape of B.1.1.7 and Victoria (VIC) severe acute respiratory syndrome coronavirus 2 (SARS-CoV-2) strains. (**A**) Experimental design to compare replication kinetics and transcriptomic landscapes of VIC and B.1.1.7 strains. A549-ACE2 cells were seeded and 24 hr later inoculated with VIC or B.1.1.7 strain (multiplicity of infection [MOI] = 1) for 2 hr. Non-internalised viruses were removed by trypsin digestion, and cells were fixed at designated timepoints for single-molecule fluorescence in situ hybridisation (smFISH) or harvested for RNA-seq library preparation. (**B**) Maximum

*Figure 6 continued on next page*

*Figure 6 continued*

z-projected confocal images of A549-ACE2 cells infected with VIC or B.1.1.7. Representative super-permissive cells from the time series are shown. Numbers at the bottom-left corner indicate dynamic contrast range used to display the image. Scale bar = 10 μm. (**C**) Comparison of the percentage of super-permissive cells between the two strains. Super-permissive cells were identified from low-magnification high-throughput smFISH assay (see *Figure 6—figure supplement 1B-C*). Data are represented as mean ± SD (8 hr post infection [hpi] n = 2; 24 hpi n = 3). *p<0.5: **p<0.01. (**D**) Read coverage along SARS-CoV-2 genome (positive strand) for the two variants in the three timepoints. Counts are normalised to total read count to show the increased proportion of reads from the virus in addition to the accumulation of subgenomic RNA and averaged across replicates. (n = 3). (**E**) Percentage of reads mapping to SARS-CoV-2 genome of total mapped reads, shown separately for the two strains. Each symbol represents an experimental replicate (n = 3). (**F**) Violin plots showing fold-changes in the host transcriptome and viral RNA genome comparing B.1.1.7 and VIC strains at the three timepoints. Fold-changes for SARS-CoV-2-positive and -negative strands are indicated as separate points and coloured according to the statistical significance of the change (red – higher in VIC, blue – higher in B.1.1.7, grey – no change). p-adjusted <0.01, log2 fold-change cut-off = 0 (n = 3). (**G**) Percent of reads mapping to SARS-CoV-2-negative (antisense) strand relative to all SARS-CoV-2 reads, shown separately for the two strains. Each symbol represents an experimental replicate (n = 3). (**H**) Estimated ratio of SARS-CoV-2 subgenomic to genomic RNA for the two virus variants at the three timepoints. Student's *t*-test. n.s., not significant; *p<0.05; ****p<0.001 (n = 3). (**I**) Expression of S, N, ORF9b, and N* viral subgenomic RNAs in each strain and different timepoints. Expression of each subgenomic RNA is determined from split reads indicative of transcriptional skipping landing within 100 nt upstream of annotated ORF start site, or until upstream ORF start codon if nearer. Percentage of all skip events is shown (n = 3).

The online version of this article includes the following figure supplement(s) for figure 6:

**Figure supplement 1.** Delayed replication kinetics of B.1.1.7 severe acute respiratory syndrome coronavirus 2 (SARS-CoV-2) variant in A549-ACE2 cells.

**Figure supplement 2.** Transcriptomic landscapes of B.1.1.7 and Victoria (VIC) and severe acute respiratory syndrome coronavirus 2 (SARS-CoV-2) strains.

## Transcriptomic changes in B.1.1.7 and VIC-infected cells

To further explore the differences in gene expression between the B.1.1.7 and VIC strains, we assessed the abundance of the different vRNAs in infected A549-ACE2 cells. Negative sense viral RNAs represent a small fraction of the vRNA present in the cell, as assayed by smFISH (*Figure 6G*). These negative sense transcripts are detectable as early as 2 hpi, adding further support to our earlier conclusion that primary viral replication events can occur rapidly post-infection, particularly in 'super-permissive' cells (*Figures 3C and 6G*). The ratio between negative and positive sense vRNAs increased throughout the infection for the VIC strain, but for B.1.1.7 we observed a modest reduction in the ratio at 24 hpi (*Figure 6G*). To assess the expression of sgRNAs, we quantified the reads mapping to the split junctions derived from RNA-dependent RNA polymerase discontinuous replication (*Figure 6—figure supplement 2C*; *Kim et al., 2020*; *V'kovski et al., 2021*). In agreement with smFISH data, sgRNAs were detected in low quantities at 2 hpi (*Figure 6D and G*). For VIC, the sgRNA/gRNA ratio peaks at 8 hpi, followed by a significant drop at 24 hpi (*Figure 6G*). For B.1.1.7, we observed a significantly lower sgRNA/gRNA ratio at 8 hpi when compared to VIC (*Figure 5H*). However, the sgRNA/gRNA ratio of B.1.1.7 remained stable between 8 and 24 hpi, surpassing VIC (*Figure 6H*). These results suggest that both VIC and B.1.1.7 have a different kinetics of gRNA and sgRNA expression, complementing our earlier observations with smFISH (*Figure 3C and G*).

Next, we assessed the relative abundance of each individual sgRNA. We found that S-sgRNA was the dominant species at 2 hpi, while the sgRNA encoding N (N-sgRNA) become prevalent at 8–24 hpi (*Figure 6—figure supplement 2D*). We interpret this early S-to-N sgRNA switch upon infection as indicating a transition to the assembly of viral particles requiring large numbers of N molecules. At 2 hpi, B.1.1.7 produced more S-sgRNA but less N-sgRNA than VIC, which is consistent with a delayed B.1.1.7 replication kinetic and S-to-N transition. Furthermore, we found upregulation of sgRNAs encoding ORF9b (~0.13%) and N* (~1%) in B.1.1.7-infected cells (*Figure 6I*), in agreement with recent studies reporting altered sgRNA landscapes for B.1.1.7 (*Parker et al., 2021*; *Thorne et al., 2021*). Upregulation of these transcripts is likely to result from advantageous mutations that create novel transcriptional regulatory sequences (TRS-B) in B.1.1.7 (*Parker et al., 2021*; *Wang et al., 2021*). When we scanned for the TRS motifs in other VOCs and variants of interests (VOIs), we found mutations in TRS-B near N* were also found in P.1 (Gamma) and P.2 (Zeta) variants while mutations in TRS-B near ORF9b were unique to B.1.1.7 (see *Supplementary file 1*). However, multiple sequence alignment of VOCs and VOIs revealed that mutations accumulate frequently at or near the TRS motif sequences, suggesting that SARS-CoV-2 utilises these regulatory motif and surrounding sequences as evolutionary hotspots to modulate sgRNA expression and viral fitness. These transcriptomic results reveal that B.1.1.7 does not only exhibit delayed replication kinetics, but also produces a differential

pool of sgRNAs likely due to mutations within the TRS. Altogether, the novel combination of smFISH and 'in-bulk' RNA sequencing that we have described provides a powerful and holistic way to characterise the replication dynamics of SARS-CoV-2. Our pipeline can now be expanded to other VOC, viruses, functional analyses, and characterisation of antivirals.

## Discussion

Our spatial quantitation of SARS-CoV-2 replication dynamics at the single-molecule and single-cell level provides important new insights into the early rate-limiting steps of infection. Typically, analyses of viral replication are carried out using 'in-bulk' approaches such as RT-qPCR and conventional RNA-seq. While very informative, these approaches lack spatial information and do not allow single-cell analyses. Although single-cell RNA-seq analyses can overcome some of these issues (*Fiege et al., 2021*; *Ravindra et al., 2021*), their low coverage and lack of information regarding the spatial location of cells remain a significant limitation. In this study, we show that smFISH is a sensitive approach that allows the absolute quantification of SARS-CoV-2 RNAs at single-molecule resolution. Our experiments show the detection of individual gRNA molecules within the first 2 hr of infection, which most likely reflect incoming viral particles. However, we also observed small numbers of foci comprising several gRNAs sensitive to RDV treatment, demonstrating early replication events. We believe that these foci represent 'replication factories' as they co-stain with FISH probes specific for negative sense viral RNA and sgRNA. These data provide the first evidence that SARS-CoV-2 replication occurs within the first 2 hr of infection and increases over time. This contrasts to our observations with the J2 anti-dsRNA antibody where viral-dependent signals were apparent at 6 hpi (*Cortese et al., 2020*; *Eymieux et al., 2021*). We noted that co-staining SARS-CoV-2-infected cells with J2 antibody and +ORF1a with an smFISH probe set showed a partial overlap, suggesting that infection may induce changes in cellular dsRNA. These findings highlight the utility of smFISH to uncover new aspects of SARS-CoV-2 replication that are worthy of further study.

We found that SARS-CoV-2 gRNA persisted in the presence of RDV, suggesting a long half-life that may reflect the high secondary structure of the RNA genome that could render it refractory to the action of nucleases (*Huston et al., 2021*; *Simmonds et al., 2021*). smFISH revealed complex dynamics of gRNA and sgRNA expression that resulted in a rapid expansion of sgRNA (peaking at 8 hpi), followed by a shift towards the production of gRNA (24 hpi), results that were confirmed by RNA-seq. Since a viral particle is composed of thousands of proteins and a single RNA molecule, we interpret the high synthesis of sgRNAs as aiming to fulfil the high demand for structural proteins in the viral particles. Once the structural proteins are available in sufficient quantities, the late shift towards gRNA synthesis may ensure the presence of sufficient gRNA to generate the viral progeny.

Our study shows that cells vary in their susceptibility to SARS-CoV-2 infection, where most cells had low vRNA levels (<$10^2$ copies/cell), but a minor population (4–10% depending on the cell line) had much higher vRNA burden (>$10^5$ copies/cell) at 10 hpi. In contrast, the number of intracellular vRNA copies at 2 hpi was similar across the culture, suggesting that this phenotype is not explained by differences in virus entry. These 'super-permissive' cells account for the majority of vRNA within the culture and mask the dominant cell population. Similar results were obtained with Vero E6, Calu-3, and A549-ACE2, suggesting that this is a common feature of SARS-CoV-2 infection. As both Calu-3 and A549-ACE2 have intact innate sensing pathways (*Cao et al., 2021*; *Li et al., 2021*) unlike Vero cells (*Desmyter et al., 1968*), this variable susceptibility is unlikely to reflect differential immune cell signalling and is consistent with their random distribution within the culture. The reason for the differential infection fitness may rely on the intrinsic properties of each cell, including the stage of the cell cycle, the expression of individual antiviral sensors or the metabolic state. Recent single-cell RNA sequencing studies of SARS-CoV-2-infected bronchial cultures identified ciliated cells as the primary target. However, only a minority of these cells contained vRNA that may either reflect low sequencing depth or cell-to-cell variation in susceptibility (*Fiege et al., 2021*; *Ravindra et al., 2021*), and indeed, our smFISH results on experimentally infected hamster lungs showed variable RNA levels across the tissue (*Figure 1E*). The human respiratory tract encompasses the nasal passage, large and small airways and bronchioles, and our knowledge on how specific cell types and SARS-CoV-2 RNA burden relate is still limited. Applying smFISH in concert with cell-type-specific labels to clinical biopsies and experimentally infected animal samples (*Salguero et al., 2021*) will allow us to address this important question.

Given the current status of the pandemic, there has been a global effort to understand the biology of emergent VOC with high transmission rates and possible resistance to neutralising antibodies. Most studies have focused on mutations mapping to the Spike glycoprotein as they can alter virus attachment, entry, and sensitivity to vaccine-induced or naturally acquired neutralising antibodies. However, many of the mutations map to other viral proteins, including components of the RNA-dependent RNA polymerase complex that could impact RNA replication, and non-coding regulatory regions as the TRSs, which can affect sgRNA expression. Our smFISH analysis revealed that the B.1.1.7 variant shows slower replication kinetics compared to the VIC strain, resulting in lower gRNA and sgRNA copies per cell, fewer viral replication factories and a reduced frequency of 'super-permissive' cells. This delay in B.1.1.7 replication was observed in Vero and A549-ACE2 cells and was confirmed by RNA-seq as an orthogonal method.

Emerging VOCs, such as B.1.1.7, have been reported to have a fitness advantage in terms of their ability to transmit compared to the VIC isolate (*Caly et al., 2020*; *Davies et al., 2021*; *Kidd et al., 2021*; *Volz et al., 2021*). However, the mechanisms underlying increased transmission are not well understood. Interestingly, a recent study reported that B.1.1.7 leads to higher levels of intracellular vRNA and N protein than VIC at 24 and 48 hpi using 'in-bulk' RT-qPCR and immunofluorescence, respectively (*Thorne et al., 2021*). We observed that while B.1.1.7 still produces lower level of vRNA than VIC at 24 hpi (*Figure 6E*), it exhibits a clear recovery compared to 8 hpi. It is thus plausible that both variants yield similar total amounts of viral RNA and proteins but within a different time frame. The potential differences in replication dynamics between the two variants are also reflected in distinct sgRNA/gRNA ratios throughout the infection (*Figure 6H*). That said, 'in-bulk' RT-qPCR analysis does not provide absolute quantification and individual cell assessment and, therefore, should likely be biased towards the super-susceptible cells that account for most of the RNA burden. Thorne and colleagues also reported an elevated expression of the sgRNA encoding the innate agonist ORF9b (*Thorne et al., 2021*), which is also supported by our results. We noticed that the increase of ORF9b sgRNA expression may be due to mutations in non-coding regulatory sequences involved in discontinuous replication (TRS), and that such mutations are common across VOCs possibly mediating differential sgRNA expression. Enhanced ORF9b expression, together with the lower intracellular vRNA levels present in B.1.1.7-infected cells, may grant this variant with an advantage to evade the antiviral response. This advantage, combined with mutations in the Spike that are proposed to improve cell entry, could provide the B.1.1.7 with a replicative advantage over the early lineage VIC strain enabling its rapid dissemination across the human population (*Caly et al., 2020*; *Davies et al., 2021*; *Kidd et al., 2021*; *Volz et al., 2021*). A recent longitudinal study of nasopharyngeal swabs showed that the B.1.1.7 variant was associated with longer infection times and yet showed similar peak viral loads to non-B.1.1.7 variants (*Kissler et al., 2021*). The authors conclude that this extended duration of virus shedding may contribute to increased transmissibility and is consistent with our data showing reduced replication of B.1.1.7 at the single-cell level. An independent study reported a significantly longer duration of SARS-CoV-2 RNA in nasopharyngeal swabs from persons infected with B.1.1.7 (16 days) compared to those infected with other lineages (14 days) (*Calistri et al., 2021*). Replication fitness will be defined by the relationship of the virus with its host cell, and aggressive replication is expected to trigger cellular antiviral sensors. In contrast, lower replication may allow the virus to replicate and persist for longer periods before host antiviral sensors are triggered. Such differences, and their impact on host antiviral responses, are likely to be of key importance for our understanding of the success of viral variants to spread through the population.

## Materials and methods

### Key resources table

| Reagent type (species) or resource | Designation | Source or reference | Identifiers | Additional information |
|---|---|---|---|---|
| Gene (SARS-CoV-2) | SARS-CoV-2 RefSeq reference genome | NCBI | NC_045512.2 | |
| Gene (*Homo sapiens*) | human genome | Ensembl | GRCh38.99 | |

*Continued on next page*

*Continued*

| Reagent type (species) or resource | Designation | Source or reference | Identifiers | Additional information |
|---|---|---|---|---|
| Strain, strain background (SARS-CoV-2) | Victoria 01/20 (BVIC01) | *Caly et al., 2020* | | Provided by PHE Porton Down after supply from the Doherty Centre Melbourne, Australia |
| Strain, strain background (SARS-CoV-2) | B.1.1.7 (20I/501Y.V1.HMPP1) | *Tegally et al., 2020* | | Provided by PHE Porton Down |
| Strain, strain background (SARS-CoV-2) | HCoV-229E | Provided by Professor Andrew Davidson and Professor Peter Simmonds | | |
| Cell line (African green monkey) | Vero E6 | Kind gift from Professor William James | | |
| Cell line (*H. sapiens*) | A549-ACE2 | Kind gift from Professor Ralf Bartenschlager | | |
| Cell line (*H. sapiens*) | Huh-7.5 | Kind gift from Professor Peter Simmonds | | |
| Cell line (*H. sapiens*) | Calu-3 | Kind gift from Professor Nicole Zitzmann | | |
| Biological sample (Gold Syrian hamster) | hamster lung tissue | This study | | Infected tissue provided by the Biological Investigations Group, Public Health England, Porton Down |
| Antibody | J2 primary antibody (mouse, monoclonal) | Scicons | Cat# 10010200; RRID:AB_2651015 | 1:500 IF |
| Antibody | Anti-SARS-CoV-2 N antibody (human, monoclonal) | *Huang et al., 2020* | Ey2A clone | 1:2000 IF |
| Commercial assay or kit | SARS primer probe | IDT | Cat# 100006770 | |
| Commercial assay or kit | B2M primer probe | Applied Biosystems | Cat# 4325797 | |
| Commercial assay or kit | QIAGEN RNeasy kit | QIAGEN | Cat# 74004 | |
| Commercial assay or kit | Illumina Total RNA Prep with Ribo-Zero Plus library kit | Illumina | Cat# 20040525 | |
| Commercial assay or kit | Beckman Coulter RNAClean XP beads | Beckman Coulter | Cat# A63987 | |
| Commercial assay or kit | Takyon Dry Probe MasterMix | Eurogentec | Cat# UFD-NPMT-C0101 | |
| Chemical compound, drug | ATTO633 NHS ester | Atto-Tec | Cat# AD633 | |
| Chemical compound, drug | ATTO565 NHS ester | Atto-Tec | Cat# AD565 | |
| Chemical compound, drug | Cy3 NHS ester | Lumiprobe | Cat# 11320 | |
| Chemical compound, drug | ATTO488 NHS ester | Atto-Tec | Cat# AD488 | |
| Chemical compound, drug | Phalloidin-Alexa Fluor 488 conjugate | Thermo Fisher | Cat# A12379 | |
| Chemical compound, drug | CellMask Green | Thermo Fisher | Cat# C37608 | |

*Continued on next page*

*Continued*

| Reagent type (species) or resource | Designation | Source or reference | Identifiers | Additional information |
|---|---|---|---|---|
| Chemical compound, drug | RNaseT1 | Thermo Fisher | Cat# EN0541 | |
| Chemical compound, drug | RNaseIII | NEB | Cat# M0245S | |
| Software, algorithm | Stellaris Probe Designer ver 4.2 | Biosearch technologies | | |
| Software, algorithm | 'bowtie2' | *Langmead and Salzberg, 2012* | | v2.4.4 |
| Software, algorithm | OMERO server | *Allan et al., 2012* | | |
| Software, algorithm | cellSens Dimension | Olympus | | |
| Software, algorithm | ImageJ | National Institute of Health | | |
| Software, algorithm | FISH-quant | *Mueller et al., 2013* | | v3 |
| Software, algorithm | RNA Distribution Index (RDI) calculator | *Stueland et al., 2019* | | |
| Software, algorithm | STAR aligner | *Dobin et al., 2013* | | v2.7.3a |
| Software, algorithm | Cellpose | *Stringer et al., 2021* | | v0.6.1 |
| Software, algorithm | Bigfish | *Imbert et al., 2021* | | v0.4.0 |
| Software, algorithm | R package DESeq2 | *Love et al., 2014* | | v1.28.1 |
| Software, algorithm | Affinity Designer | Serif | | |

## Cell culture

Vero E6, A549-ACE2 (kind gift from the Bartenschlager lab) (*Klein et al., 2020*), and Huh-7.5 cells were maintained in standard DMEM, Calu-3 cells in Advanced DMEM both supplemented with 10% foetal bovine serum, 2 mM L-glutamine, 100 U/ml penicillin, and 10 µg/ml streptomycin and non-essential amino acids. All cell lines tested free of mycoplasma were maintained at 37°C and 5% $CO_2$ in a standard culture incubator. Human cell line identities were authenticated by Short Tandem Repeat (STR) profiling, and Vero E6 cells were verified using DNA barcoding.

## Virus propagation and infection of cell culture models

*SARS-CoV-2 strains*: VIC 01/20 (BVIC01) (*Caly et al., 2020*) (provided by PHE Porton Down after supply from the Doherty Centre Melbourne, Australia) and B.1.1.7 (*Tegally et al., 2020*) (20I/501Y. V1.HMPP1) (provided by PHE Porton Down). Viral strains were propagated in Vero E6 cells as described (*Wing et al., 2021*). Briefly, naïve Vero E6 cells were infected with SARS-CoV-2 at an MOI of 0.003 and incubated for 48–72 hr until visible cytopathic effect was observed. At this point, cultures were harvested, clarified by centrifugation to remove residual cell debris and stored at –80°C. To determine the viral titre, fresh Vero E6 cells were inoculated with serial dilutions of SARS-CoV-2 viral stocks for 2 hr followed by addition of a semi-solid overlay consisting of 1.5% carboxymethyl cellulose (Sigma). Cells were incubated for 72 hr and visible plaques enumerated by fixing cells using amido black stain to calculate PFU/ml. Similarly, HCoV-229E (Andrew Davidson lab [Bristol] and Peter Simmonds lab [Oxford]) virus was propagated in Vero E6 cells and TCID50 was performed in Huh-7.5 cells.

For smFISH experiments with the SARS-CoV-2 stains, cells were infected at an MOI of 1 for 2 hr followed by extensive washing in PBS. Residual cell surface-associated virus was removed by trypsin treatment of the cell monolayer for 2 min followed by neutralisation of the trypsin using serum-containing media. Infected cells were then maintained for defined periods up to 24 hr. For the HCoV-229E, cells were infected at an MOI of 1 and were maintained for 24 and 48 hr.

## Hamster infection and tissues preparation

Golden Syrian hamsters (*Mesocricetus auratus*) (males and females) aged 7 weeks old, weighing 96–116 g, were obtained from Envigo, London, UK. Hamsters were housed in individual cages with

access to food and water ad libitum. Hamsters were briefly anaesthetised with 5% isoflurane (Zoetis, Leatherhead, UK) and 4 l/m $O_2$ and inoculated by the intranasal route with $5 \times 10^4$ PFU/animal of SARS-CoV-2 BVIC01 delivered in 100 µl per nostril (200 µl in total). Hamsters were monitored post-infection for weight, clinical signs and temperature (via implanted temperature chip). On day 4, the hamsters were euthanised by overdose (sodium pentobarbitone [Dolelethal, Vetquinol UK Ltd]) via the intraperitoneal route. At necropsy, lung samples were fixed in 10% buffered formalin at room temperature and embedded in paraffin wax. 4 µm tissue sections were cut.

## RT-qPCR
Infected cells were harvested in RLT buffer and RNA extracted using the QIAGEN RNeasy kit. SARS-CoV-2 RNA was quantified using a one-step reverse transcriptase qPCR (RT-qPCR) kit (Takyon) in a multiplexed reaction containing primer probes directed against the SARS-CoV-2 N gene (FAM) and ß–2-microglobulin (VIC) as an internal control. All qPCR reactions were carried out using a Roche 96 Light cycler (Roche) (SARS primer probe IDT CAT:100006770, B2M primer probe Applied Biosystems 4325797).

## Single-molecule fluorescence in situ hybridisation (smFISH)
smFISH was carried out as previously reported (*Titlow et al., 2018*; *Yang et al., 2017*) with minor modifications. Briefly, cells were grown on #1.5 round-glass coverslips in 24-well plate or in µ-slides 8-well glass bottom (IBIDI) and fixed in 4% paraformaldehyde (Thermo Fisher) for 30 min at room temperature. Coverslips were cleaned in 80% ethanol with lint-free tissue and kept in 100% ethanol to maintain sterility and cleanliness. Cells were permeabilised in PBS/0.1% Triton X-100 for 10 min at room temperature followed by washes in PBS and 2× SSC. Cells were pre-hybridised in pre-warmed (37°C) wash solution (2× SSC, 10% formamide) twice for 20 min each at 37°C. Hybridisation was carried out in hybridisation solution (2× SSC, 10% formamide, 10% dextran sulphate) containing 500 nM smFISH probes overnight at 37°C. For infection timepoints beyond 24 hr, smFISH probes were added at 1 µM. After the overnight hybridisation, cells were washed for 20 min in pre-warmed wash solution at 37°C followed by counterstain with DAPI (1 µg/ml), phalloidin-Alexa Fluor 488 conjugate (264 nM) and/or CellMask Green (1:1,000,000) diluted in wash solution. Cells were then washed once with wash solution for 20 min at 37°C and twice with 2× SSC for 10 min each at room temperature. Cells were mounted using Vectashield, IBIDI mounting media or 2× SSC.

For RNase digestion experiments, RNaseT1 (Thermo Fisher, EN0541, 100 U/ml) or RNaseIII (M0245S, NEB, 20 U/ml) was used to degrade ssRNA and dsRNA, respectively. Permeabilised cells were treated with RNases in PBS supplemented with 5 mM $MgCl_2$ and incubated at 37°C for 1 hr and washed three times with PBS.

In the experiment to detect viral negative strands, dsRNA was denatured using DMSO, formamide, or NaOH (*Singer et al., 2021*; *Wilcox et al., 2019*). After the permeabilisation step, cells were rinsed in distilled water and were treated with 50mM NaOH for 30s at room temperature, 70% formamide at 70°C for 1hr, or 90% DMSO at 70°C for 1hr. Following the treatments, cells were quickly cooled on ice, washed in ice-cold PBS and subjected to standard smFISH protocol. The smFISH experiments in *Figures 3 and 5* were performed with DMSO and heat denaturation.

For smFISH on FFPE hamster lungs, the tissue sections were pre-treated as described in *Annaratone et al., 2017* and the probes were hybridised based on the protocol described in *Rouhanifard et al., 2018* with minor modifications. Briefly, tissues were fixed in 10% neutral buffered formalin and sectioned to 5µm slices. Tissue sections were deparaffinised in xylene (2 × 10min), washed in 100% ethanol (2 × 5min) and post-fixed in methanol-acetic acid (3:1v/v) for 5min. Tissues were re-hydrated in an ethanol gradient for 3min each (100%, 85%, 70%, nuclease-free water), heated at 80°C for 1hr in antigen retrieval solution (10mM sodium citrate, pH 6 supplemented with 1:50 RVC), permeabilised in 70% ethanol overnight at 4°C. Then, sections were incubated in 100% ethanol for 5min, air-dried for 5min and tissue-cleared with 8% SDS made up in 2× SSC. Afterwards, standard smFISH procedures were followed.

## smFISH probe design and specificity analysis
Candidate smFISH probe sequences were acquired using Stellaris Probe Designer version 4.2 (https://www.biosearchtech.com/stellaris-designer) with the following parameters: organism, human; masking

level, 5; oligo length, 20 nt; minimum spacing length, 3 nt. Appropriate region of the SARS-CoV-2 Wuhan-Hu-1 (NC_045512.2) reference sequence was used as target sequence. We BLAST screened candidate probe sequences against custom human transcriptome and intron database to score number of off-target basepair matches, then 35–48 sequences with the least match scores were chosen per probe set. Oligonucleotides were singly labelled with ATTO633, ATTO565, Cy3, or ATTO488 at 3' ends according to a published protocol (*Gaspar et al., 2017*) and were concentration normalised to 25µM. All probe sets used in this study had degree of labelling>0.94.

We developed a bespoke pipeline to analysed the sequence specificity of oligonucleotide probe sequences against ORF-1a and ORF-N by alignment against SARS-CoV-1 (NC_004718), SARS-CoV-2 (NC_045512), MERS-CoV (NC_019843), HCoV-229E (NC_002645), HCoV-NL63 (NC_005831), HCoV-OC43 (NC_006213), HCoV-HKU1 (NC_006577), human (GCF_000001405.39), and African green monkey (GCF_015252025.1) RefSeq genome or transcriptome assembly using 'bowtie2' (2.4.4) (*Langmead and Salzberg, 2012*). Following bowtie2 arguments were used to find minimum edit distance of oligonucleotide sequences to target genome/transcriptome: `--end-to-end --no-unal --align-seed-mm 0`, `--align-seed-length 5`, `--align-seed-interval 1-1.15`, `--effort-extend 15`, `--effort-repeat 2`. Melting temperatures were obtained using 'rmelting' (1.8.0) R package at 300mM Na concentration (2× SSC). smFISH probe sequences used in this study are available in *Supplementary file 2*.

## Immunofluorescence

After permeabilisation, cells were blocked in blocking solution (50% LI-COR Odyssey blocking solution, pretreated with RNASecure for 30 min and supplemented with 2 mM ribonucleoside vanadyl complex and 0.1% Tween-20) for 30 min at room temperature. Then, cells were incubated with J2 primary antibody (Scicons 10010200) at 2 µg/ml or human anti-N primary antibody (Ey2A clone 1:2000) (*Huang et al., 2020*) for 2 hr at room temperature. Cells were washed three times in PBS/0.1% Tween-20 (PBSTw) for 10 min each at room temperature and incubated with fluorescent secondary antibodies (1:500) diluted in blocking solution for 1 hr at room temperature. After further three washes in PBSTw, cells were mounted using Vectashield or IBIDI mounting media. For combined smFISH and immunofluorescence, antibody staining was carried out sequentially after the smFISH protocol.

## Microscopy and image handling

Cells were imaged on an Olympus SpinSR10 spinning disk confocal system equipped with Prime BSI and Prime 95B sCMOS cameras. Objectives used were ×20 dry (0.8 NA, UPLXAPO20X), ×60 silicone oil (1.3 NA, UPLSAPO60XS2), ×60 oil (1.5 NA, UPLAPOHR60X), or ×100 oil (1.45 NA, UPLX-APO100XO). Image voxel sizes were 0.55 × 0.55 × 2 µm (x:y:z) with the ×20 objective and 0.11 × 0.11 × 0.2 µm (x:y:z) with the ×60 and ×100 objectives. Automatic and manual image acquisition and image stitching were performed with Olympus cellSens Dimension software. Images were uploaded and stored in the University of Oxford OMERO server (*Allan et al., 2012*), and OMERO.figure (3.2.0) was used to generate presented image visualisations.

## Image analysis
### Cell segmentation and counting

Cell segmentation was performed either manually in ImageJ (National Institute of Health) or automatically with Cellpose (0.6.1) (*Stringer et al., 2021*) using 2D maximum intensity projected images of phalloidin or CellMask stains. Cellpose parameters for ×60 and ×100 magnification images were model_type = cyto, diameter = 375, flow_threshold = 0.9, cellprob_threshold=-3. For 20× stitched images, CellMask channel was deconvolved with constrained iterative module using cellSens (five iterations, default spinning disk PSF, Olympus), then the following Cellpose parameters were used: model_type = cyto, diameter = 55, flow_threshold = 0, cellprob_threshold=-6. Total number of cells per image was counted using a custom ImageJ macro script or from the Cellpose segmentation output on DAPI channel images (model_type = nuclei, diameter = 20, default threshold). Infected cells were counted using ImageJ '3D object counter' or manually.

## Quantification of smFISH images

Single-molecule-level quantification of smFISH images was performed either with FISH-quant (**Mueller et al., 2013**) or Bigfish (**Imbert et al., 2021**). For FISH-quant, ImageJ region of interest (ROI) files were converted to FQ outline file using a custom Python script. Then, smFISH channels were Laplacian of Gaussian filtered (sigma = 7, 3 px) and pre-detected using local maximum mode with 'allow smaller z region for analysis' option enabled. Pre-detected diffraction limited spots were fitted with 3D Gaussian and thresholded in batch mode based on filtered intensity, amplitude, and σz. Thresholds were defined by uninfected 'Mock' condition samples. The filtering also removed non-specific autofluorescence and rare dust particles because these contaminants usually show lower fluorescence intensity and are highly variable in shapes.

Large smFISH datasets were processed with a custom Python pipeline using Bigfish, skimage, and numpy libraries (available in the GitHub repository). Tif files were converted to a numpy array, and individual cells were segmented from the image using the Cellpose library as described above. Images where cells were labelled with the CellMask stain were pre-processed with a median filter, radius = 50. Background signal in the smFISH channel was subtracted with the skimage.white_tophat algorithm (radius = 5, individual z frames were processed in 2D due to memory constraints, results were indistinguishable from 3D-processed images). Threshold setting for smFISH spot detection was set specifically for each set of images collected in each session.

Cells with high viral RNA ($>10^{5-6}$ RNA counts) were quantified by integrating smFISH channel intensities within entire cellular volumes and comparing to the reference integrated intensity of single molecules derived from cells with lower infection density. Reference single-molecule images were obtained using 'Average spots' in TxSite mode of FISH-quant or 'build_reference_spot()' function in Bigfish.

Viral factories were defined as gRNA smFISH signals with spatially extended foci that exceed the point-spread function of the microscope and intensity of the reference single molecules. In FISH-quant, the foci were quantified using the TxSite quantification mode (xy:z = 500:1200 nm crop per factory) with normal-sampled averaged single-molecule image (xy:z = 15:12 px) from the batch mode output. Then, 'Integrated intensity in 3D' method was used to compare integrated intensity of the viral factory to that of averaged single-molecule RNA. In Bigfish, the factories were resolved using 'decompose_cluster()' function to find a reference single-molecule spot in a less signal-dense region of the image, which was used to simulate fitting of reference single-molecule spots into viral factories until the local signal intensities are matched. The candidate factories were filtered based on the previously reported radii of DMVs measured by electron microscopy (150 nm pre-8 hpi and 200 nm post-8 hpi) (**Cortese et al., 2020**). In addition, we applied a threshold of 3–7 RNA molecules per factory as a technical cut-off to prevent overestimation or over-cluster of viral factories at later infection timepoints.

## Dual-colour smFISH spot detection analysis

The same viral RNA target was detected using two smFISH probes labelled with alternating (ODD and EVEN) red and far-red fluorochromes. Resulting images were processed in FISH-quant to obtain 3D coordinates of each spots. Percentage co-localisation analysis was performed with a custom script using an R package 'FNN' (1.1.3). Briefly, we calculated 3D distance of nearest neighbour for each spot in the red channel to the closest detected spot in the other channel and repeated the analysis starting from the far-red channel. We then used a value of 300 nm to define co-localised spots corresponding to the same viral RNA molecule. The presented visuals report percentage co-localisations calculated from the red channel to the far-red channel and vice versa. The analysis was performed per field of view.

## Quantification of fluorescence intensity and signal co-localisation

Immunofluorescence images were background subtracted using rolling ball subtraction method (radius = 150 px) in ImageJ. Anti-dsRNA (J2) stain was quantified by integrating fluorescence signal across the z-stacks of cellular ROI divided by the cell volume to obtain signal density. Signal density was normalised to the average signal density of uninfected 'Mock' condition cells. Fluorescence intensity profiles were obtained using ImageJ 'plot profile' tool across 3 μm region on 1 μm maximum intensity projected images. To assess co-localisation of N protein with SARS-CoV-2 RNA, ellipsoid mask centred

around centroid xyz coordinates of smFISH spots was generated with the size of the point-spread function (xy radius = 65 nm, z radius = 150 nm) using ImageJ 3D suite. Integrated density of N protein channel (background subtracted, radius = 5 px) fluorescence within the ellipsoid mask was measured and compared to the equivalent signal in the uninfected condition or randomly distributed ellipsoids.

## Calculation of RNA spatial dispersion index

Subcellular spatial distribution metrics of SARS-CoV-2 RNA species were quantified using the RNA Distribution Index (RDI) calculator (*Stueland et al., 2019*). Nuclei and cell boundaries were pre-segmented in ImageJ using the 'Auto Threshold' function on DAPI ('method = Huang') or CellMask green ('method = MaxEntropy') channels. Resulting images were maximum intensity projected and passed into the RDI calculator MATLAB script. Standard RDI calculator graphical user interface was used without background intensity removals.

## Simulation of super-permissive cell distribution

Simulations were performed to determine if the appearance of SARS-CoV-2 super-permissive cells follows a random distribution. The general strategy was to test the complete spatial randomness hypothesis by comparing the average nearest-neighbour distance of superinfected cells to an equal number of randomly selected coordinates (*Ripley, 1979*). 2D spatial coordinates of superinfected cells were obtained from the 3D object counter (ImageJ) as described above. Cell nuclei were segmented with the DAPI channel, and placement of random coordinates was confined to pixels that fell within the DAPI segmentation mask. Nearest-neighbour distances were calculated using the KDtree algorithm (*Maneewongvatana and Mount, 1999*) implemented in Python (scipy.spatial.KDTree). Pseudo-random distributions were simulated by randomly placing the first coordinate, then constraining the placement of subsequent coordinates to within a defined number of pixels. $Rn$ nearest-neighbour statics (*Pinder and Witherick, 1972*) were calculated according to the following equation, where D(obs) is the average nearest-neighbour distance (μm), a is the total imaged area (μm), and n is the number of super-permissive cells. $Rn$ value of 1 suggests a random distribution, whereas $Rn < 1$ indicates clusteredness and $Rn > 1$ indicates regular distributions.

$$Rn = \frac{D(obs)}{0.5\sqrt{\frac{a}{n}}}$$

## RNA-sequencing library preparation

RNA from infected cells were extracted as described above. Sequencing libraries were prepared using the Illumina Total RNA Prep with Ribo-Zero Plus library kit (Cat# 20040525) according to the manufacturer's guidelines. Briefly, 100 ng of total RNA was first depleted of the abundant ribosomal RNA present in the samples by rRNA-targeted DNA probe capture followed by enzymatic digestion. Samples were then purified by Beckman Coulter RNAClean XP beads (Cat# A63987). Obtained rRNA-depleted RNA was fragmented, reverse transcribed, converted to dsDNA, end repaired, and A-tailed. The A-tailed DNA fragments were ligated to anchors allowing for PCR amplification with Illumina unique dual indexing primers (Cat# 20040553). Libraries were pooled in equimolar concentrations and sequenced on Illumina NextSeq 500 and NextSeq 550 sequencers using high-output cartridges (Cat# 20024907), generating single 150-nt-long reads.

## RNA-sequencing analysis

### Genomes

We downloaded the human genome primary assembly and annotation from Ensembl (GRCh38.99) and the SARS-CoV-2 RefSeq reference genome from NCBI (NC_045512.2). We combined the human and viral genome and annotation files into one composite genome and annotation file for downstream analyses.

### Alignment and gene counts

We performed a splice-site-aware mapping of the sequencing reads to the combined human and SARS-CoV-2 genome and annotation using STAR aligner (2.7.3a) (*Dobin et al., 2013*). We also used

STAR to assign uniquely mapping reads in strand-specific fashion to the Ensembl human gene annotation and the two SARS-CoV-2 strains.

## Principal component analysis

To assess if SARS2 infection is the main driver of differences in the RNA-seq samples, we performed a principal component analysis (PCA). First, we performed library size correction and variance stabilisation with regularised-logarithm transformation implemented in DESeq2 (1.28.1) (*Love et al., 2014*). This corrects for the fact that in RNA-seq data variance grows with the mean, and therefore, without suitable correction, only the most highly expressed genes drive the clustering. We then used the 500 genes showing the highest variance to perform PCA using the prcomp function implemented in the base R package stats (4.0.2).

## Differential expression analysis

We performed differential expression analysis using the R package DESeq2 (1.28.1) (*Love et al., 2014*). DESeq2 estimates variance-mean dependence in count data from high-throughput sequencing data and tests for differential expression based on a model using the negative binomial distribution. Full output of DESeq2 analysis is available in *Supplementary file 3*.

## SARS-CoV-2 subgenomic RNA expression

To assess relative levels of viral subgenomic and genomic RNA expression, we tallied the alignments (using GenomicRanges and GenomicAlignments R packages; *Lawrence et al., 2013*) mapping to the region unique to the genomic RNA and the shared region and normalised for their respective lengths. Unique contribution of sgRNA region was then estimated by subtracting the contribution of the genomic RNA from the shared region. In order to assess expression of individual SARS-CoV-2 subgenomic RNAs, we extracted split (junction) reads mapping to the viral genome with the GenomicAlignments R package (1.24.0) (*Lawrence et al., 2013*). The subgenomic transcripts fully overlap the full genomic RNA and partially with each other. While the molecular process generating these subgenomic RNAs is distinct from RNA splicing, from the point of view of short read mapping they are equivalent. We determined the relative expression level of each sgRNA generated by transcriptional skipping by calculating the number of reads supporting skipping into a region upstream of each annotated viral ORF. To avoid spurious mappings, we filtered for skip sites that were present in all three replicates and constituted at least 0.1% of all skipped viral reads.

## Statistics, data wrangling, and visualisation

Statistical analyses were performed in R (3.6.3) and RStudio (1.4) environment using an R package 'rstatix' (0.7.0). p-Values were adjusted using the Bonferroni method for multiple comparisons. The 'tidyverse suite' (1.3.0) was used in R, and 'Numpy' and 'Pandas' Python packages were used in Jupyter notebook for data wrangling. The following R packages were used to create the presented visualisation: 'ggplot2' (3.3.2), 'ggbeeswarm' (0.6.0), 'hrbrthemes' (0.8.0), 'scales' (1.1.1), and 'patchwork' (1.1.1). Further visual annotations were made in the Affinity Designer (Serif).

## Acknowledgements

We are grateful to Danail Stoychev and Maria Kiourlappou for the advice on Python programming and high-performance computing. We are very grateful to Olympus UK and Europe for their generous loan of an Olympus IXplore SpinSR spinning disk system for the imaging work in this project and to Matthew Freeman and Jordan Raff for enabling us to install the microscope in the Dunn School of Pathology specifically for this SARS-CoV-2 work. We thank Michael Knight, Maeva Dupont, Lisa Chauveau, and Javier Gilbert Jaramillo for their provision of resources and assistance in Category III containment labs. We thank Micron Advanced Imaging Unit (https://micronoxford.com) for provision of advanced microscopy facilities and technical advice. ID: The Davis laboratory is funded by a Wellcome Investigator Award 209412/Z/17/Z and Wellcome Strategic Awards (Micron Oxford) 091911/B/10/Z and 107457/Z/15/Z. JAM: The McKeating laboratory is funded by a Wellcome Investigator Award 200838/Z/16/Z, UK Medical Research Council (MRC) project grant MR/R022011/1, and the Chinese Academy of Medical Sciences (CAMS) Innovation Fund for Medical Science (CIFMS), China (grant

number: 2018-I2M-2–002). AC is supported by an MRC Career Development Award (MR/L019434/1), MRC grants (MR/R021562/1, MC_UU_12014/10, and MC_UU_12014/12), and John Fell funds, University of Oxford. FJS is funded by the UK Health Security Agency (UK-HSA). James & Lillian Martin Centre is generously supported by the James Martin 21st Century Foundation. MKT is funded by a Leverhulme Grant to ID. AW and RMP are supported by Wellcome Strategic Award 107457/Z/15/Z. WJ, MP-B and TB are funded by the COVID-19 Research Response Fund, University of Oxford. JYL and DSG are funded by the Medial Sciences Graduate Studentship, University of Oxford.

## Additional information

### Funding

| Funder | Grant reference number | Author |
| --- | --- | --- |
| Wellcome Trust | Investigator Award 209412/Z/17/Z | Jeffrey Y Lee<br>Dalia S Gala<br>Aino I Järvelin<br>Mary Kay Thompson<br>Richard M Parton<br>Ilan Davis<br>Joshua Titlow |
| Medical Research Council | MC_UU_12014/12 | Marko Noerenberg<br>Alfredo Castello |
| Wellcome Trust | Strategic Award 091911/B/10/Z | Alan Wainman<br>Ilan Davis |
| Wellcome Trust | Strategic Award 107457/Z/15/Z | Alan Wainman<br>Ilan Davis |
| Wellcome Trust | Investigator Award 200838/Z/16/Z | Peter AC Wing<br>Xiaodong Zhuang<br>Jane A McKeating |
| Medical Research Council | MR/R022011/1 | Peter AC Wing<br>Xiaodong Zhuang<br>Jane A McKeating |
| Chinese Academy of Medical Sciences | 2018-I2M-2-002 | Peter AC Wing<br>Xiaodong Zhuang<br>Jane A McKeating |
| Medical Research Council | Career Development Award MR/L019434/1 | Marko Noerenberg<br>Natasha Palmalux<br>Louisa Iselin<br>Alfredo Castello |
| Medical Research Council | MR/R021562/1 | Marko Noerenberg<br>Natasha Palmalux<br>Louisa Iselin<br>Alfredo Castello |
| Medical Research Council | MC_UU_12014/10 | Marko Noerenberg<br>Natasha Palmalux<br>Louisa Iselin<br>Alfredo Castello |
| University of Oxford | Medical Sciences Graduate Studentship | Jeffrey Y Lee<br>Dalia S Gala |
| University of Oxford | COVID-19 Research Response Fund | William James<br>Maria Prange-Barczynska<br>Tammie Bishop |
| James Martin 21st Century Foundation | James & Lillian Martin Centre | William James |
| University of Oxford | John Fell funds | Alfredo Castello |
| UK Health Security Agency | | Francisco J Salguero |

| Funder | Grant reference number | Author |
| --- | --- | --- |
| Leverhulme Trust | | Mary Kay Thompson<br>Ilan Davis |

The funders had no role in study design, data collection and interpretation, or the decision to submit the work for publication.

## Author contributions

Jeffrey Y Lee, Conceptualization, Data curation, Formal analysis, Investigation, Methodology, Project administration, Software, Visualization, Writing – original draft, Writing – review and editing; Peter AC Wing, Conceptualization, Data curation, Formal analysis, Investigation, Methodology, Project administration, Resources, Visualization, Writing – original draft, Writing – review and editing; Dalia S Gala, Data curation, Formal analysis, Investigation, Methodology, Visualization, Writing – review and editing; Marko Noerenberg, Conceptualization, Formal analysis, Methodology, Resources; Aino I Järvelin, Data curation, Formal analysis, Investigation, Software; Joshua Titlow, Formal analysis, Investigation, Methodology, Software; Xiaodong Zhuang, Natasha Palmalux, Louisa Iselin, Investigation, Methodology; Mary Kay Thompson, Investigation, Methodology, Software; Richard M Parton, Investigation, Methodology, Resources; Maria Prange-Barczynska, Francisco J Salguero, Investigation, Resources; Alan Wainman, Methodology, Resources; Tammie Bishop, William James, Resources; Daniel Agranoff, Conceptualization, Project administration, Supervision; Alfredo Castello, Conceptualization, Funding acquisition, Investigation, Methodology, Project administration, Supervision, Writing – original draft, Writing – review and editing; Jane A McKeating, Ilan Davis, Conceptualization, Funding acquisition, Investigation, Methodology, Project administration, Resources, Supervision, Writing – original draft, Writing – review and editing

## Author ORCIDs

Jeffrey Y Lee http://orcid.org/0000-0002-5146-0037
Peter AC Wing http://orcid.org/0000-0002-2354-3281
Dalia S Gala http://orcid.org/0000-0001-5089-7504
Marko Noerenberg http://orcid.org/0000-0002-4834-8888
Aino I Järvelin http://orcid.org/0000-0002-1225-4396
Joshua Titlow http://orcid.org/0000-0002-9586-4797
Richard M Parton http://orcid.org/0000-0002-2152-4271
Alan Wainman http://orcid.org/0000-0002-6292-4183
Daniel Agranoff http://orcid.org/0000-0002-9246-0308
William James http://orcid.org/0000-0002-2506-1198
Alfredo Castello http://orcid.org/0000-0002-1499-4662
Jane A McKeating http://orcid.org/0000-0002-7229-5886
Ilan Davis http://orcid.org/0000-0002-5385-3053

## Ethics

Animal experimental work was conducted under the authority of a UK Home Office approved project license that had been subject to local ethical review at UK Health Security Agency, Porton Down by the Animal Welfare and Ethical Review Body (AWERB) as required by the 'Home Office Animals (Scientific Procedures) Act 1986'.

## Decision letter and Author response

Decision letter https://doi.org/10.7554/eLife.74153.sa1
Author response https://doi.org/10.7554/eLife.74153.sa2

# Additional files

## Supplementary files

• Supplementary file 1. Multiple sequence alignment of severe acute respiratory syndrome coronavirus 2 (SARS-CoV-2) variants of concern and variants of interest centred around transcriptional regulatory sequence motifs.

• Supplementary file 2. Single-molecule fluorescence in situ hybridisation (smFISH) oligonucleotide probe sequences used in this study.

• Supplementary file 3. Differentially expressed gene analysis of RNA sequencing samples from A549-ACE2 cells infected with B.1.1.7 or Victoria SARS-CoV-2 strains at 2, 8, and 24 hr post infection.

• Transparent reporting form

### Data availability

The presented RNA sequencing data has been deposited at Gene Expression Omnibus (GEO), with accession number GSE184932. Source codes for reproducing figures and smFISH image analysis pipeline scripts are available at the GitHub repository: https://github.com/jefflee1103/Lee_Wing-SARS2 (copy archived at swh:1:rev:bc3e724233d321bf5599979061ffbe0cb907da03).

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
