## [Editor Report]

This manuscript uses imaging and analysis techniques to study the kinetics and diversity of RNA production in SARS-CoV-2 infected cells. This presents a novel insight into the biology and life cycle of the virus, and this knowledge has the potential to inform future interventions for COVID-19.

---

## [Decision Letter]

[Editors' note: this paper was reviewed by Review Commons.]

**Decision letter after peer review:**

Thank you for resubmitting your work entitled "Absolute quantitation of individual SARS-CoV-2 RNA molecules provides a new paradigm for infection dynamics and variant differences" for further consideration by *eLife*. Your revised article has been reviewed by 1 peer reviewer and the evaluation has been overseen by Miles Davenport as the Senior Editor and Reviewing Editor.

The manuscript has been improved but there are some remaining issues that need to be addressed, as outlined below:

The reviewer has identified a number of issues where further discussion would improve the clarity and impact of the manuscript. In particular, a discussion of the issues in (1) and inclusion of existing additional data on hamster physiology (further experiments are not required), as well of discussion of issues (2) and (3). I am optimistic this can be rapidly completed.

*Reviewer #2:*

In this revised manuscript the authors have done well to justify their methodologies in their application of smFISH to quantify early-stage replication kinetics of two SARS-CoV-2 strains in several model systems. While the outcomes of this research are valuable and important, several critiques are outlined below.

1. The authors have adapted an already established technique, smFISH, to thoroughly explore early infection kinetics of two important and distinct SARS-CoV-2 stains. The authors are able to visualize infection at much earlier time points than can be detected by other techniques, which is valuable for distinguishing strain-specific differences in replication. However, the authors focus a significant portion of their manuscript on assessing and classifying cells based on how permissive they are to viral replication following the earliest stages of infection. While these variations are interesting, the authors have not sufficiently justified the value of these data to a biologically relevant system. The inclusion of Golden Syrian hamster data is appreciated, but the authors have not correlated replication heterogeneity with cell type or any other factor that might explain why certain cells are more permissive than others. Additionally, the authors should include any physiological data (weight, clinical signs, and temperature), acquired during these experiments in this manuscript. Above all, it is not surprising to observe heterogeneous tissue infection patterns by smFISH or any other similar technique as a totally uniform infection would likely be catastrophic to the host and disadvantageous for viral transmission. If the authors propose that variations in these infection patterns are important, these findings should be justified with some data to support mechanism.

2. The delayed infection kinetics of the B.1.1.7 variant are very interesting and potentially have important clinical and public health implications. Is there any correlation between slow early-stage infection kinetics and potentially longer asymptomatic/presymptomatic infection in patients that could help explain the increased transmissibility of this variant?

3. In Figure 1, panel B, the authors demonstrate smFISH with viruses immobilized on a coverslip. The possibility exists that some background fluorescence could be derived from dust or other contamination, so a negative control should be included with these images.

---

## [Author Response]

The reviewer has identified a number of issues where further discussion would improve the clarity and impact of the manuscript. In particular, a discussion of the issues in (1) and inclusion of existing additional data on hamster physiology (further experiments are not required), as well of discussion of issues (2) and (3). I am optimistic this can be rapidly completed.Reviewer #2:In this revised manuscript the authors have done well to justify their methodologies in their application of smFISH to quantify early-stage replication kinetics of two SARS-CoV-2 strains in several model systems. While the outcomes of this research are valuable and important, several critiques are outlined below.1. The authors have adapted an already established technique, smFISH, to thoroughly explore early infection kinetics of two important and distinct SARS-CoV-2 stains. The authors are able to visualize infection at much earlier time points than can be detected by other techniques, which is valuable for distinguishing strain-specific differences in replication. However, the authors focus a significant portion of their manuscript on assessing and classifying cells based on how permissive they are to viral replication following the earliest stages of infection. While these variations are interesting, the authors have not sufficiently justified the value of these data to a biologically relevant system. The inclusion of Golden Syrian hamster data is appreciated, but the authors have not correlated replication heterogeneity with cell type or any other factor that might explain why certain cells are more permissive than others. Additionally, the authors should include any physiological data (weight, clinical signs, and temperature), acquired during these experiments in this manuscript. Above all, it is not surprising to observe heterogeneous tissue infection patterns by smFISH or any other similar technique as a totally uniform infection would likely be catastrophic to the host and disadvantageous for viral transmission. If the authors propose that variations in these infection patterns are important, these findings should be justified with some data to support mechanism.

To address these points, we have now added viral parameters and physiological data (weight, clinical signs and temperature) on the infected hamsters in the revised paper [Page 5, line 31]. In depth quantification of smFISH signals in the representative infected lung tissue and the nature of infected cells is ongoing and is beyond the remit of the current paper. Our rationale for including this data was to show that the smFISH technique can be applied to fixed tissue samples to highlight its potential use in clinical studies. We have revised the text slightly to make sure this point is clear [Page 6, line 7] and [Page 16, lines 13-18].

2. The delayed infection kinetics of the B.1.1.7 variant are very interesting and potentially have important clinical and public health implications. Is there any correlation between slow early-stage infection kinetics and potentially longer asymptomatic/presymptomatic infection in patients that could help explain the increased transmissibility of this variant?

This is a good point made by the reviewer. In addition to the study by Kissler 2021 that reported longer infection times with B.1.1.7 variant following repeated daily sampling, we now also discuss an additional report (Calistri 2021). This study shows longer duration of PCR detectable viral RNA, albeit with lower Ct values, in nasopharyngeal swabs from B.1.1.7 variants compared to those infected with other lineages. The results from this study are consistent with extended replication times in the upper respiratory tract associating with increased transmissibility. To address the referee comment, we have revised the manuscript text to include these points [Page 17, line 25].

3. In Figure 1, panel B, the authors demonstrate smFISH with viruses immobilized on a coverslip. The possibility exists that some background fluorescence could be derived from dust or other contamination, so a negative control should be included with these images.

To address this point, we have now included negative control images from the same experiment in [Figure 1—figure supplement 1B] where poly-L-lysine coated coverslips were incubated in PBS without the SARS-CoV-2 virus. These control images are free of signal within the intensity range we expect for single RNA molecules detected by smFISH probes, consistent with the fact that we clean all the glassware very carefully with ethanol before cell seeding and imaging. To further support our interpretation of the fluorescent spots we observe being immobilised virus particles containing viral RNA, we have also included a density plot showing a unimodal distribution of spot fluorescence intensities that are similar to spots we detect within cells in [Figure 1—figure supplement 1B]. In our experience, signals from dust particles and other contaminants are shaped very differently from smFISH RNA spots, which have a typical sub-diffraction point spread function, characteristic of the objective lens and optical path of the microscope. We have revised the texts to clarify our experimental and image processing methods [Page 35, line 27] and [Page 38, line 21].

[Editors' note: we include below the reviews that the authors received from Review Commons, along with the authors’ responses.]

Reviewer #1:Major comments:The authors used approaches provided in FISH-quant (Mueller et al., Nat Methods 2013) and big-fish. However, these tools to analyze RNA aggregates were not designed and validated for such massive aggregations as observed by SARS-Cov-2. They were developed for cases such as transcription sites with much smaller aggregations, with a few tens to a hundred molecules. With a regular spot detection approach, usually a few thousand spots can be detected in a cell (e.g. King et al., J Virol 2018), but this depends also on the used microscope and the available cellular volume. Higher RNA concentrations cannot be resolved with a standard approach, because RNA spots start to overlap. Decomposing RNA aggregations can help but will not work reliably for the high RNA densities observed for SARS-Cov-2, especially at later infection time-points. The tools will then not provide accurate estimates anymore. To my knowledge, there is currently not accurate quantification method for such massive RNA levels in smFISH. What has been done in the past, is using cellular intensity as an approximation and perform calibrations with cells having lower and thus still resolvable RNA counts (Raj et al., PLO Biology; https://doi.org/10.1371/journal.pbio.0040309.sg003). The authors proposed three expression regimes (partially resistant, permissive, and super permissive). My concerns here apply mainly to the category super-permissive, where an accurate estimation can't be performed. Here a more cautious quantification should be applied. To a lesser extent, this will also apply to some of quantifications of gRNAs per factory, with counts exceeding 100s of molecules. As mentioned above, this does not affect any of the conclusions, but would reflect more accurately what kind of reliable information can be drawn from such experiments.

We agree with the reviewer that approaches like FISH-quant and Big-FISH are most sensitive when applied to sparse single molecule images. These methods cannot be used accurately to quantify RNA spots with high spatial density of single molecules, such as our examples of “super-permissive” cells. In fact, single molecule quantitation of such high-density cases is likely to underestimate RNA expression as noted by us and King et al. 2018 (doi: 10.1128/JVI.02241-17). Therefore, we integrated the combined smFISH signal intensity within entire cellular volumes and compared to the mean intensity of single molecules in cells with lower infection density. To address this comment, we have (i) revised the methods and Results sections to explain more carefully and explicitly the quantification of RNA in super-permissive cells [page 8, line 31; page 38, line 30] and (ii) provided a calibration plot (Figure 3—figure supplement 1) for the quantitation as previously reported (Raj et al. 2006, doi: 10.1371/journal.pbio.0040309).

We agree that high local RNA density has the potential to interfere with quantification of gRNAs within viral factories (the last part of the comment above). To address this point, we have also revised the methods section of the manuscript to explain more fully that we have used the “cluster.decomposition()” function of Big-FISH to quantify viral factories [page 39, line 1]. This method of analysis is conceptually similar to the “Integrated intensity” mode of FISH-quant. Applying this algorithm to non-super permissive cells allows us to use the me an intensity of a reference single-molecule spot to estimate the number of molecules in a cluster. We are confident such estimates are reliable in the majority of viral factories, which contain less than or equal to 200 single gRNA molecules.

Reviewer #1:Minor comments:1. Page 6; the authors state that "smFISH identifies.… cellular distribution.…. within ER-like membranous structures". However, the authors didn't directly show such a localization, could they provide an experiment with an ER stain?

The text on page 6 that the reviewer is referring to was based on previous light microscopy and EM studies that reported SARS-CoV-2 RNA in ER-derived membranes (termed Double Membrane Vesicles – DMVs) or co-localisation of anti-dsRNA (J2) with ER-markers (Cortese et al. 2020; Hackstadt et al. 202; Mendonca et al. 2021). To address this comment, we have clarified the text and cited these publications as well as toned down our claim that the virus is located in ER-like membranous structures [page 6, line 20].

Cortese et al. 2020, doi: 10.1016/j.chom.2020.11.003

Hackstadt et al. 2021, doi: 10.3390/v13091798

Mendonca et al. 2021, doi: 10.1038/s41467-021-24887-y

2. It might be worthwhile pointing out that the probe-sets can be used in different host organisms (Vero – African green monkey; human cell lines).

We have revised the text to emphasise more clearly the applicability of SARS-CoV-2 probes for the study of many different host organisms [page 6, line 1]. Namely, human, monkey and now hamster.

3. I really liked the experiment, where the authors showed absence of signal when infecting with another virus and elegant control with the J2 AB. Maybe the authors could explain more clearly that the used a different coronavirus and that based on their sequence alignment no/little signal would be expected.

Thank you for this supportive comment. We followed the reviewer’s suggestion and expanded our explanation of the rationale of this experiment in the text [page 5, line 18].

4. I might have missed this, but they authors could also mention the positive control data about but Calu3 infected with SARS-COv2. One thing I was wondering: why did the authors use two different cell lines for this experiment?

To address this good point, we have added a sentence about a positive control visualising SARS-CoV-2 in Calu-3 cells using our probe set [page 5, line 25].

The experiments with HCoV-229E were done in Huh-7.5 cells because SARS-CoV-2 and HCoV-229E have distinct cell preferences. Using the J2 antibody we show that the levels of the dsRNA derived from viral replication are similar in the two cell lines and with the two viruses. Therefore, the lack of smFISH signal in HCoV-229E infected cells strongly supports the high specificity of the probe set.

5. Figure 1E. Would be nice to have the intensity scale for all time-points to permit a comparison of image intensities along the different time-points.6. Figure 3B. Would be important to have intensity scale bars to judge the signal intensities across the different time-points.

The fluorescence intensity scale in Figure 1E is applicable to all timepoints, except for the lower panel at 24 hpi, which was intended to show wider dynamic contrast range. To address this point and avoid possible confusion, we have provided intensity scales for all time-points studied in this figure and also in Figure 3B.

7. The experiment with the isolated virions shows nicely that the smFISH approach has single-virus sensitivity. Did the authors compare the intensity of these isolated virions with the signal in Figure 1B? This might be a question of personal taste, but to me, this section might actually fit better in the first paragraph of page 4/5, where the authors describe single virions in cells.

Thank you for the interesting question. We have not performed a direct comparison of the spot intensities of intracellular genomic RNA molecules and those from the isolated virions, because the two samples are not identical in their fluorescence background. While, isolated SARS-CoV-2 requires poly-L-lysine coating of the coverslip (which introduces background fluorescence), our infection strategy utilises cells (which have their own background fluorescence) growing on uncoated glass. Nonetheless, the isolated virion spot intensities follow a unimodal distribution, and their shape approximates to the point-spread function of the microscope. Since spots at 2 hpi are largely derived from non-replicative viral genomes and they are measured in the intracellular environment with the same background (autofluorescence), they represent a better ‘single RNA molecule’ reference. We have revised the Methods section to briefly explain how we obtained reference single molecules in our smFISH analysis [page 38, line 33; page 39, line 6].

We also thank the reviewer for suggesting rearranging the text section. We have moved the relevant text to the second paragraph of the Results section [page 5, line 9], as they suggested.

8. Page 6. The authors state "+ORF-N and +ORF-S single labelled spots, corresponding to sgRNAs, were more uniformly distributed throughout the cytoplasm than dual labelled gRNA". This is difficult to appreciate from the image. Is this something the authors could quantify, e.g. with the metrics proposed by Stueland et al., Scientific Reports 2019?

To address this point, we have: (i) presented an alternative image with cell and nuclear boundaries illustrating a clearer example of differential spatial localisation of gRNA and sgRNA (Figure 2B), and (ii) performed quantification of spatial dispersion indices for gRNA and sgRNA using the suggested method for our revision (Figure 2—figure supplement 1). These new quantifications of the results are consistent with our observation that sgRNA are dispersed throughout the cytoplasm but gRNAs are rather confined to perinuclear regions [page 6, line 35].

9. Page 6. The authors perform a FISH/IF experiment including a co-localization analysis, where a "limited overlap" with sgRNAs was observed. I was wondering if this overlap could actually be simply due to rather high density of the sgRNAs. Maybe a control analysis by slightly changing the RNA positions could provide insight here, and give a threshold for what's to be expected randomly at a given RNA density.

The reviewer’s comment is correct, in that a high density of sub genomic RNAs (sgRNAs) and nucleocapsid protein could lead to signal overlap due to chance. This is why we excluded “super-permissive” cells from this analysis. Our co-localisation data shows that genomic RNA (gRNA) spots have a bimodal nucleocapsid immunofluorescence intensity distribution, consistent with the existence of two populations of gRNAs: one that is nucleocapsid-associated and one “free”. Nevertheless, we agree with the reviewer that the analysis of randomly positioned transcripts of the same density provides a valuable control and clarifies how we defined the threshold. To address this point, in our revised manuscript we have included: (i) a random distribution analysis comparing the overlap between sgRNA/gRNA and nucleocapsid in the “Observed” and “Randomised” simulations [page 7, line 7], and (ii) a plot showing a full distribution of co-localised nucleocapsid immunofluorescence intensity for both genomic and sub-genomic viral RNAs (Figure 2—figure supplement 2).

10. I don't fully follow the argument about stability on page 8. The authors also see an increase in the RNA levels. Couldn't this increase compensate for loss of RNA due to degradation? Would it be possible to perform an experiment at a very high REMDESIVIR concentrations which would blocks transcription?

Remdesivir (RDV) is a nucleoside analogue that inhibits viral RNA polymerase activity**.** While this drug inhibits most viral replication, the inhibition is incomplete and using higher concentrations results in cellular toxicity. At the present time there are no stronger polymerase inhibitors available, so these experiments offer the best estimation currently possible to assess viral RNA stability. Estimations of RNA stability in any biological context are by their very nature always only approximations. For example, similar experiments in mammalian cells where transcription is inhibited with flavopiridol, DRB or actinomycin D also include some degree of inhibition escape. We thank the referee for highlighting this limitation that we now address by revising the text [page 9, line 14].

11. Figure 3C. maybe indicate the two groups with dashed lines.

We have added a dashed line at the 10^2^ mark in Figure 3C to visually differentiate “partially resistant” and “permissive” cells.

12. How did the authors define/detect replication factories? I couldn't find information about this in the methods.

This is a good point raised by both the reviewers. Please see [Reviewer 2 General comment #1] for our response.

Reviewer #2:General comments:1. The authors' definition of viral factories, in part as foci with at least 4 gRNA molecules, comes across as arbitrary. Perhaps a clearer explanation of this cutoff would be helpful to the readers' understanding of this definition. Additionally, confirmation of the functionality of such factories by immunofluorescence with anti-RdRp, for example, in addition to identifying staining of gRNAs and (-) sense viral RNAs at each focus could provide valuable support to the authors' conclusions.

We thank both reviewers for requesting further information regarding our explanation of viral factories. We defined viral factories as smFISH signals with spatially extended foci that exceed the size of the point spread function of the microscope (representing how a diffraction limited spot appears in the image) and the intensity of a reference single molecule. We then filtered these candidate factories based on the radius of the signal foci with EM-measured radii of double-membrane vesicles and single-membrane vesicles formed by SARS-CoV-2 (150 nm pre-8hpi and 200 nm post-8hpi), as previously defined in the literature (Cortese et al. 2020; Mendoca et al. 2021). Our terminology encompasses both replication and viral assembly sites. The threshold of 4 genomic RNA molecules was selected as a technical threshold to limit an over-estimation of viral factories at later timepoints. For our spinning-disk confocal imaging system, we found the threshold of 3-7 RNA molecules provided satisfactory results. To address both referee comments, we revised the Results and Methods sections to clarify our rationale for defining and quantifying viral factories [page 10, line 17; page 39, line 1].

As the reviewer mentioned, we have shown a partial overlap of positive sense genomic RNAs with negative sense genomic RNAs (Figure 2D, Figure 2—figure supplement 3), suggesting these viral factories represent double membrane vesicles. The use of antibodies against the viral polymerase (nsp12) is also a possibility to detect replication centres. However, replication centres are not the only ‘viral factories’ as there are also double-membrane structures where viral particles assemble (Mendoca et al. 2021) and they, in principle, lack negative sense RNA and replication machinery, so neither smFISH probes against the negative strand nor a nsp12 antibody will comprehensively detect viral factories. We appreciate the valuable suggestion, but the classification of viral factories into replication and assembly sites would be very challenging due to reagent availability and is beyond the scope of this manuscript.

2. The random distribution of super-permissive cells in each cell line was demonstrated early in the infection, primarily at 8 hpi. The authors do not show how this pattern changes over time (8, 10, 12, 16, 24 hpi, for example). Do clusters of super-permissive cells appear at later time points, or does the pattern of 'highly' infected cells remain random for each virus? Any strain-specific differences identified from such patterns may be important for understanding infection progression. Finally, the authors do acknowledge this point, but it cannot be overstated that these data were taken from cell culture systems that have limited similarities to the human respiratory epithelium. A better model for such studies might be primary cultured human bronchial epithelial cells, but of course, these cells are not as readily accessible as the cell lines used in this manuscript.

We share the reviewer’s view that the presence and the spatial distribution of “super-permissive” cells can provide unique insights into SARS-CoV-2 infection dynamics. Our findings suggest that even at 24 hours post infection (hpi), not all cells become “super-permissive” and the culture maintains a heterogenous population of “partially resistant”, “permissive” and “super-permissive” cells (Figure 3C, Figure 3—figure supplement 2). We also agree with the reviewer that the spatial distribution of “super-permissive” cells at later timepoints is of interest. To address this point, we have: (i) analysed the spatial distribution of “super-permissive” cells at 24 hpi, and (ii) compared the distribution of “super-permissive” cells at 24 hpi between VIC and B.1.1.7 strains (Figure 5—figure supplement 2) [page 12, line 14]. Our new analysis reveals that super-permissive cells infected with B.1.1.7 exhibit a clustered distribution at 24 hpi, in contrast to VIC strains, in which the super-permissive cells were randomly distributed.

We entirely agree with the comment on the limitations of the cell culture systems to the human respiratory tract. However, Calu-3 and A549-ACE2 lung epithelial cells have been used in many studies over the last year and we feel it is important to publish single cell quantitation with these models to enable comparison with the published literature. We believe our results provide valuable information on the intrinsic nature of host cell susceptibility to support viral replication. Nevertheless, to address this valid comment, we have now also applied our smFISH probes to detect SARS-CoV-2 RNA in infected Golden Syrian hamster lung sections, which showed an uneven distribution of infected cells. While the identification and spatial characterisation of precise susceptible cell types in the lung are beyond the scope of this manuscript, we are excited to include this data in our revised manuscript to demonstrate the utility of this sensitive approach to track spatiotemporal viral infection dynamics in an intact tissue in a living animal model [page 5, line 29].

3. The difference in early replication kinetics between the VIC and B.1.1.7 strains is an exciting finding that may have implications for clinical outcomes and transmissibility of these viruses. However, the authors did not clearly demonstrate how these differences in RNA production correlate to infectious viral load released from these cells (in bulk) at each time point. An explanation of this omission would be helpful.

To address this point, we have provided data on the level of infectious virus secreted from VIC and B.1.1.7 infected cells at all time points in the revised manuscript (Figure 6—figure supplement 1D) [page 12, line 28].

In my opinion, findings related to specific cell lines are of much less importance (and are much less biologically relevant) that identification of replicative differences among strains. Such differences could be used, in part, to aid prediction of the transmissibility of VOC, for example. I think this point gets a bit 'lost in the weeds' of the rest of the paper.

To address this comment, we emphasised the differential replication kinetics of the SARS-CoV-2 strains in Results section [page 12, line 32] and in the Impact Statement of our revised manuscript, which will hopefully draw the attention of readers from a diverse range of disciplines. We have now also added a statement, for the more general reader, that the lower replication rate of B.1.1.7 is in contrast to its more rapid spread in a human population.